# Noncovalently fused-ring electron acceptors with near-infrared absorption for high-performance organic solar cells

Hao Huang[1,5], Qingxin Guo[1,5], Shiyu Feng[1], Cai'e Zhang[1], Zhaozhao Bi[2], Wenyue Xue[2], Jinjin Yang[3], Jinsheng Song [4], Cuihong Li[1], Xinjun Xu[1], Zheng Tang[3], Wei Ma[2] & Zhishan Bo[1]

Non-fullerene fused-ring electron acceptors boost the power conversion efficiency of organic solar cells, but they suffer from high synthetic cost and low yield. Here, we show a series of low-cost noncovalently fused-ring electron acceptors, which consist of a ladder-like core locked by noncovalent sulfur–oxygen interactions and flanked by two dicyanoindanone electron-withdrawing groups. Compared with that of similar but unfused acceptor, the presence of ladder-like structure markedly broadens the absorption to the near-infrared region. In addition, the use of intramolecular noncovalent interactions avoids the tedious synthesis of covalently fused-ring structures and markedly lowers the synthetic cost. The optimized solar cells displayed an outstanding efficiency of 13.24%. More importantly, solar cells based on these acceptors demonstrate very low non-radiative energy losses. This research demonstrates that low-cost noncovalently fused-ring electron acceptors are promising to achieve high-efficiency organic solar cells.

[1] Key Laboratory of Energy Conversion and Storage Materials, College of Chemistry, Beijing Normal University, 100875 Beijing, China. [2] State Key Laboratory for Mechanical Behavior of Materials, Xi'an Jiaotong University, 710049 Xi'an, China. [3] Center for Advanced Low-dimension Materials, State Key Laboratory for Modification of Chemical Fibers and Polymer Materials, College of Materials Science and Engineering, Donghua University, 201620 Shanghai, China. [4] Engineering Research Center for Nanomaterials, Henan University, 475004 Kaifeng, China. [5]These authors contributed equally: Hao Huang, Qingxin Guo. Correspondence and requests for materials should be addressed to X.X. (email: xuxj@bnu.edu.cn) or to Z.T. (email: ztang@dhu.edu.cn) or to W.M. (email: msewma@mail.xjtu.edu.cn) or to Z.B. (email: zsbo@bnu.edu.cn)

Fused-ring electron acceptors (FREAs) for use in organic solar cells (OSCs) have received extensive attention in recent years because of their easy structural modulation[1–3], broad and intense optical absorption[4], and impressive power conversion efficiency (PCE), which has exceeded 14%[5,6]. Generally, high-efficiency electron acceptors possess an acceptor–donor–acceptor (A-D-A) structure with a central planar ladder-type fused-ring D unit[7,8]. Such a planar feature is critical for the delocalization of π-electrons. Zhan and colleagues[9–13] demonstrated that FREAs with a central D unit of up to 11 fused rings are beneficial to achieve high PCEs. However, the synthesis of such large fused-ring structures is difficult and expensive[14–16]. In our previous work[17], we have developed a kind of noncovalently fused-ring electron acceptors (NC-FREAs), in which the intramolecular noncovalent interactions can lock the aromatic units to form a ladder-like structure. Similar to covalent fused-ring acceptors[18], such noncovalently fused-ring acceptors also possess broad absorption, high electron mobility, and reasonable power conversion efficiency. Importantly, the synthesis of such noncovalently fused-ring electron acceptors is easier and cheaper than that of covalent acceptors.

Herein, we design and synthesized four NC-FREAs (DOC6-IC, DOC8-IC, DOC2C6-IC, and DOC2C6-2F) with a central ladder-like core and two dicyanoindanone electron-withdrawing groups. As shown in Fig. 1, the ladder-like core is composed of a central 2,5-bis(alkyloxy)phenylene unit flanked by two 4,4-bis(2-ethyl-hexyl)-4H-cyclopenta[2,1-b:3,4-b']dithiophene units. The S⋯O interactions, which have been used in developing high-performance organic semiconductors by Watson and colleagues[19–21], could endow the central donor core with a planar ladder-like structure. The structure of such NC-FREAs is somewhat similar to acceptors with nine covalently fused rings, but the synthesis of the former is much easier than that of the latter. Importantly, the morphology of blend films is easily optimized by changing the central 2,5-bis(alkyloxy)phenylene unit to achieve high-efficiency OSCs. As expected, the four NC-FREAs display broad absorption bands ranging from 550 to 900 nm, whereas a similar acceptor molecule (DC6-IC) with a 2,5-dihexylphenylene unit instead of a 2,5-bis(alkyloxyphenylene) unit only absorbs light from 550 to 750 nm. Solar cells with PBDB-T:DOC-IC (DOC6-IC, DOC8-IC, and DOC2C6-IC) as the active layer display PCEs of 9.4% to 11.10%, and very low non-radiative recombination of energy losses (in the range of 0.20 to 0.22 eV). For solar cells with DC6-IC, the highest PCE is only 6.87% and the non-radiative recombination of energy loss is 0.30 eV. An as-cast solar cell with DOC2C6-IC, which bears two 2-ethylhexyl chains on the central phenyl unit, achieves a high PCE of 11.10%. The PCE is further improved to 13.24% for solar cell based on fluorinated acceptor DOC2C6-2F after thermal annealing at 100 °C for 5 min.

## Results

### Design, synthesis, and characterization of the small-molecule acceptors.
The synthetic route to the five acceptors is depicted in Fig. 1. Starting from commercially available 4,4-bis(2-ethylhexyl)-4H-cyclopenta[2,1-b:3,4-b']dithiophene (1), the abstraction of one α-proton with n-butyllithium afforded the mono anion intermediate, which was quenched with trimethylstannyl chloride to furnish compound 2 in a yield of 96%. Stille cross-coupling of two equivalents of 2 with dibromides 3a, 3b, 3c, and 3d using Pd (PPh₃)₄ as the catalyst precursor afforded compounds 4a, 4b, 4c, and 4d in yields of 90%, 80%, 67%, and 61%, respectively. The treatment of compounds 4a, 4b, 4c, and 4d with n-butyllithium produced the dianion intermediates, which were converted to dialdehydes 5a, 5b, 5c, and 5d in yields of 86%, 69%, 87%, and

70%, respectively, by quenching with dimethylformamide (DMF). Knoevenagel condensation of the dialdehydes with 2-(3-oxo-2,3-dihydro-1H-inden-1-ylidene)malononitrile afforded the target acceptor molecules (DOC6-IC, DOC8-IC, DOC2C6-IC, and DC6-IC) in yields of 57 to 93%. The reaction of 5c with 2-(5, 6-difluoro-3-oxo-2,3-dihydro-1H-inden-1-ylidene)malononitrile afforded DOC2C6-2F in a yield of 63%. Synthetic and characterization details for all compounds are provided in the Supplementary Fig. 1 and Supplementary Methods. All these small-molecule acceptors are soluble in common organic solvents such as chloroform (CHCl₃) and chlorobenzene. The thermodynamic and electrochemical properties of these acceptors are shown in Supplementary Figs. 2–4, and Supplementary Note 1 and 2.

### The optical properties of the small-molecule acceptors.
Ultraviolet–Visible (UV–Vis) absorption spectra of the five small-molecule acceptors are shown in Fig. 2. In chloroform solutions, the acceptors with a central 2,5-bis(alkyloxy)phenylene unit (DOC6-IC, DOC8-IC, and DOC2C6-IC) exhibit a broad featureless absorption ranging from 550 to 800 nm with a peak centered at ~722 nm, the fluorinated DOC2C6-2F displays a redshifted absorption with the main absorption peak located at 743 nm; whereas DC6-IC, which has a central 2,5-dihexylphenylene unit, shows a narrower featureless absorption from 500 to 700 nm with a peak located at 623 nm. In films, the four NC-FREAs with a central 2,5-bis(alkyloxy)phenylene unit display broad structured absorption ranging from 500 to 900 nm with a peak located at ~780 nm and a shoulder at ~710 nm. A DC6-IC film shows a narrow featureless absorption band from 500 to 720 nm with a peak at 634 nm. It is worth noting that large red shifts of ~56, 54, 44, and 37 nm are observed for DOC6-IC, DOC8-IC, DOC2C6-IC, and DOC2C6-2F respectively, upon moving from solution to the film state. For DC6-IC, the red shift is only 11 nm (the information is summarized in Supplementary Table 1). The appearance of an intense absorption peak at long wavelength indicates that the four NC-FREAs could form close π–π stacking interactions between their end groups in the films. This is because in the four acceptors with 2,5-bis(alkyloxy)phenylene units, the presence of intramolecular S⋯O interactions (identified by theoretical calculation in Supplementary Figs. 5 and 6 and described in Supplementary Note 3), as shown in Fig. 1b, causes the molecular backbone to adopt a planar conformation, which enhances the delocalization of π-electrons and extends the effective conjugation.

Fluorescence quantum yields ($\eta_{FL}$) of the five acceptors in CHCl₃ solutions are measured using an integrating sphere. We find that DOC6-IC, DOC8-IC, DOC2C6-IC, and DOC2C6-2F have high $\eta_{FL}$ values of 20.82%, 24.29%, 30.21%, and 20.32% respectively. Conversely, $\eta_{FL}$ of DC6-IC is only 0.49%, which is probably caused by the free rotation of its molecular backbone around the two single bonds between the two 4H-cyclopenta[2,1-b:3,4-b']dithiophene units and central 2,5-dihexylphenylene unit. The 'conformational locking" in the four NC-FREAs prevents the free rotation of the molecular backbone, which lowers the energy loss and leads to their high $\eta_{FL}$.

### The photovoltaic properties of the solar cells.
The photovoltaic properties of the acceptors are evaluated using an inverted solar cell structure of ITO/ZnO (30 nm)/active layer (70 nm)/MoO₃ (8.5 nm)/Ag (100 nm). PBDB-T is used as the electron donor materials. The current density–voltage (J–V) curves are shown in Fig. 3a. Also, photovoltaic parameters of the solar cells are summarized in Table 1 (the detailed optimization process is shown in Supplementary Tables 2–4 and described in Supplementary Note 4). The optimized solar cell containing a blend of

**Fig. 1** Molecular structure and the synthetic route of acceptors. **a** Synthetic route of DOC6-IC, DOC8-IC, DOC2C6-IC, DC6-IC, and DOC2C6-2F. **b** Chemical structures of PBDB-T and acceptors with and without intramolecular S···O interactions in the core. Reagents and conditions: (i) *n*-BuLi, THF, 0 °C; (ii) Pd(PPh₃)₄, toluene, 120 °C; (iii) *n*-BuLi, THF, −78 °C; DMF, −78 °C to r.t.; (vi) for 6a, chloroform, reflux; for 6b, chloroform, r.t.

PBDB-T and DOC6-IC (1:0.8 by weight) exhibits a PCE of 10.52% with an open-circuit voltage ($V_{oc}$) of 0.91 V, short-circuit current density ($J_{sc}$) of 19.21 mA cm$^{-2}$, and fill factor (FF) of 60.11%. The optimized solar cell based on PBDB-T: DC6-IC (1:0.8 by weight) achieves a PCE of 6.87% with a $V_{oc}$ of 0.99 V, $J_{sc}$ of 11.19 mA cm$^{-2}$, and FF of 62.21%. The higher $J_{sc}$ of the DOC6-IC solar cell than the DC6-IC one is mainly due to its broader absorption. Additional factors, leading to the improved $J_{sc}$ in the solar cells based on NC-FREAs, will be discussed later. It is also noted that the performance of the NC-FREA-based solar cell is related to the structure of lateral side chains on the central 2,5-bis(alkyloxy)phenylene linkage. The PCE of the solar cell based on PBDB-T:DOC8-IC (1:0.8 by weight) is 9.41%, with a $V_{oc}$ of 0.92 V, $J_{sc}$ of 17.74 mA cm$^{-2}$, and FF of 57.65%, while the solar cell based on PBDB-T: DOC2C6-IC (1:0.8 by weight) performed much better: we realize a PCE of 11.10%, a $V_{oc}$ of 0.93 V, $J_{sc}$ of 18.85 mA cm$^{-2}$, and FF of 63.33%. Furthermore, the replacement of the two IC terminal groups with fluorinated ones in DOC2C6-IC, solar cell based on PBDB-T:DOC2C6-2F (1:0.8 by weight) achieved an outstanding PCE of 13.24% with a much higher $J_{sc}$ of 21.35 mA cm$^{-2}$ and FF of 73.15% and decreased $V_{oc}$ of 0.85 V.

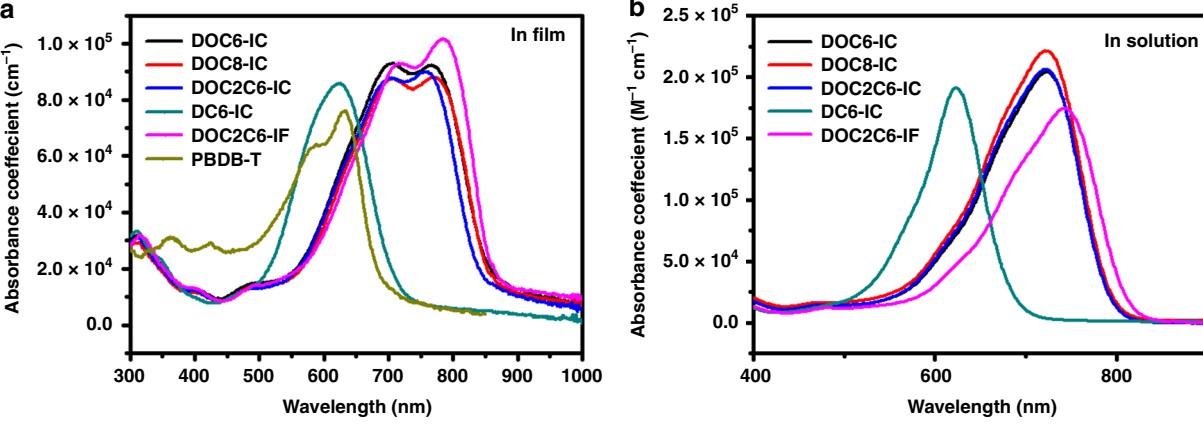

**Fig. 2** Molecular optical properties. **a** Absorption spectra of DOC6-IC, DOC8-IC, DOC2C6-IC, DC6-IC, DOC2C6-2F, and PBDB-T in films. **b** Absorption spectra of DOC6-IC, DOC8-IC, DOC2C6-IC, DC6-IC, and DOC2C6-2F in CHCl$_3$ solutions. Source data are provided as a Source Data file

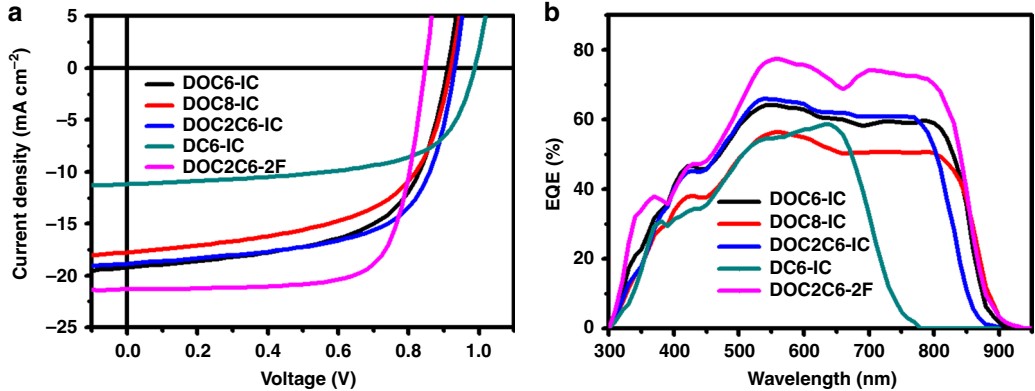

**Fig. 3** Photovoltaic performance of solar cells. **a** Current density–voltage and **b** EQE curves for DOC6-IC, DOC8-IC, DOC2C6-IC, DC6-IC and DOC2C6-2F-based OSCs. Source data are provided as a Source Data file

**Table 1 Optimal photovoltaic parameters of solar cells**

| Active layer | $V_{oc}$ [V] | $J_{sc}$ [mA cm$^{-2}$] | FF [%] | PCE$_{max}$ [%] |
|---|---|---|---|---|
| PBDB-T/DOC6-IC | 0.91 | 19.21 (17.62)[a] | 60.11 | 10.52 (10.11 ± 0.23)[b] |
| PBDB-T/DOC8-IC | 0.92 | 17.74 (15.35) | 57.65 | 9.41 (8.96 ± 0.34) |
| PBDB-T/DOC2C6-IC | 0.93 | 18.85 (16.96) | 63.33 | 11.10 (10.66 ± 0.33) |
| PBDB-T/DC6-IC | 0.99 | 11.19 (10.39) | 62.21 | 6.87 (6.77 ± 0.12) |
| PBDB-T/DOC2C6-2F | 0.85 | 21.35 (20.80) | 73.15 | 13.24 (12.99 ± 0.15) |

[a]$J_{sc}$ in parentheses is calculated by integrating the EQE spectrum with the AM1.5G spectrum
[b]Value in parentheses is the average PCE over five solar cells

External quantum efficiencies (EQEs) of solar cells are shown in Fig. 3b. The solar cell based on the similar but unfused acceptor DC6-IC has a spectral response in the wavelength range of 300–780 nm, which is in accordance with the absorption spectrum of the DC6-IC film (Fig. 2). As for NC-FREAs, all solar cells showed broad light responses ranging from around 300 to 900 nm, which are identical to the absorption spectra of blend films. The integration of EQE curves with the AM1.5 solar spectrum furnished $J_{sc}$ values that are close to those obtained by J–V measurements.

Photoluminescence (PL) quenching experiments are also employed to evaluate the domain size of each component in the blend films[22]. Fluorescence spectra of neat small-molecule acceptor films, neat PBDB-T film, and blend films are shown in Supplementary Fig. 7. Acceptor emission is efficiently quenched by PBDB-T in the blend films of DOC6-IC, DOC2C6-IC, and

DC6-IC, whereas the emission from DOC8-IC is slightly quenched in the blend film. This result indicates that the domain size of DOC8-IC is much larger than those of the former three acceptors. These PL quenching results partially explain why $J_{sc}$ and FF of DOC8-IC solar cells are the lowest among OSCs containing these NC-FREAs.

**The blend morphology of the active layers**. The morphology of the active layers is further evaluated using microscopic and X-ray diffraction methods. Atomic force microscope (AFM) images reveal that the surfaces of all blend films are highly smooth (Supplementary Fig. 8). The root-mean-square (RMS) roughness values of DOC6-IC, DOC8-IC, DOC2C6-IC, DC6-IC, and DOC2C6-2F-based blend films are 2.45, 2.55, 2.70, 3.32, and 1.16 nm, respectively. Transmission electron microscope (TEM) images suggest that there are nanoscale phase separation in all of

the blends, and formation PBDB-T fibrils with a diameter of around 16 nm for all non-fluorinated acceptors (Supplementary Fig. 8). The PBDB-T:DOC2C6-2F-based blend film shows apparently different morphologies with the diameter of the fibrils being smaller (around 10 nm)[23]. Although the molecular weight of PBDB-T has a significant influence on the morphology of active layer[24,25], the one we use here is from the same batch, which has a $M_n$ of 115,000 and a PDI of 1.63. To better understand the relationship between morphology and PCE, we further carry out resonant soft X-ray scattering (R-SoXS) experiments to analyze the morphology of active layers (Fig. 4)[26,27]. We find that the active layer of DOC8-IC shows larger domain size of 25.7 nm, while those of DOC6-IC and DOC2C6-IC exhibit smaller domain size of 22.7 nm and 15.7 nm, respectively. Owing to the limited exciton diffusion length (ca. 10 to 20 nm) in organic materials and the large domain size of DOC8-IC, the excitons generated in the acceptor domain cannot be efficiently diffused to the donor–acceptor interface. This is why DOC8-IC shows the lowest PCE among these three NC-FREAs. We use the time of flight secondary ion mass spectroscopy (ToF-SIMS) to study the vertical composition distribution in the active layer. The vertical composition distribution could be determined by the intensity of $CN^-$ signal, since the signal of $CN^-$ is only from the small molecular acceptors (the intensity of $CN^-$ for neat PBDB-T film is markedly lower than the active layers). The intensity of $CN^-$ in different depth of active layers shows no obvious change, indicating that the acceptor and donor uniformly distributed in the vertical direction (Supplementary Fig. 9).

In addition, molecular stacking in neat and blend films is investigated by grazing-incidence wide-angle X-ray scattering (GIWAXS) and the results are presented in Fig. 4, Supplementary Tables 5 and 6[28]. For the neat DOC-IC films, the π–π stacking (010) peaks in the out-of-plane (OOP) direction appear at around 1.7 Å$^{-1}$, corresponding to the π–π stacking distance of 3.7 Å, which is much shorter than that for the DC6-IC film. In addition, DOC6-IC and DOC2C6-IC tend to adopt a face-on orientation, whereas no preferred orientation is observed for DOC8-IC and DC6-IC. The intensities of the lamellar stacking peaks for the neat DOC-IC films are stronger than that of the neat DC6-IC film, indicating that DOC-ICs can form more ordered packing than DC6-IC. Moreover, the lamellar stacking peaks of DOC2C6-IC is lower than those of DOC8-IC and DOC6-IC, which may be caused by the furcated branches of DOC2C6-IC. Interestingly, the presence of (001) and (002) peaks (around 0.3 Å$^{-1}$ and around 0.57 Å$^{-1}$) of the neat DOC2C6-2F film indicates an ordered packing along the molecular backbone direction. Meanwhile the outstanding (010) peak in the OOP direction shows an enhanced face-on orientation of DOC2C6-2F. The possible driving force for such microstructural feature might be due to the introduction of fluorine atoms at the terminal groups of DOC2C6-2F, which can enhance intermolecular D–A interaction (also called as electrostatic interaction) between neighboring acceptor molecules due to the strong electron-withdrawing ability of fluorine atom[29]. This can improve the crystallization of molecules, in accordance with the result of DSC. In the blend films, the (100) scattering signals of the small molecules in DOC6-IC, DOC8-IC, and DC6-IC-based blends are emerged, which implies the ordered lamellar packing of acceptors along the in-plane or out-of-plane direction. In contrast, the (100) signal of DOC2C6-IC is absent in its blend film due to the furcate branches of side chains. Especially, the (100) signal and the (001) and (002) signals appear in DOC2C6-2F blend film, which indicates the ordered arrangement along backbone and side chain orientation of the molecule due to the introduction of the fluorine atom[29]. Furthermore, the face-on population (Fig. 4) is derived from the normalized integrated (010) peak (background corrected) as a function of polar angle

according to the reported method[30,31], giving the calculated values of 0.97, 0.95, 0.96, 0.90, and 0.98 for DOC6-IC, DOC8-IC, DOC2C6-IC, DC6-IC, and DOC2C6-2F, respectively. We can find that the DOC2C6-2F shows higher face-on orientation and the DC6-IC appears lower face-on orientation, which confirm our opinion above. Moreover, the π–π stacking peak in the DOC2C6-2F blend film along the OOP direction signifies an enhanced coherence length of 44.8 Å than the DOC2C6-IC blend film. The special orientation (an ordered packing along the molecular backbone direction) of DOC2C6-2F blend film benefits the three-dimensional transport of charge in the device, thus leading to an excellent FF[32–34]. To obtain the miscibility of active layers, we carry out the test of contact angle and calculation of Flory-Huggins interaction parameters ($\chi$) (Supplementary Note 5 and Supplementary Tables 7 and 8). It has been pointed out that a low $\chi$ value denotes a high miscibility of blend films with a more homogeneous morphology[35], which would benefit the charge transport and exciton dissociation in the active layer. DOC8-IC possesses the highest value of $\chi$ among these small molecular acceptors, namely it has the lowest miscibility with PBDB-T and the largest domain size, in accordance with the R-SoXS and PL quenching measurements.

**Quantum efficiency losses in solar cells based on NC-FREAs.** Compared to the solar cells based on similar but unfused acceptor DC6-IC, the significantly improved photocurrent extraction in the solar cells based on NC-FREAs is partially due to the boarder absorption spectra (Fig. 2). We measured the optical constants of the active materials (Fig. 5a) and performed optical transfer matrix (TM) simulations[36]. Using TM simulated active layer absorption, the maximum photocurrent extraction of the solar cells based on NC-FREAs and DC6-IC is calculated for different active layer thicknesses, by assuming 100% device internal quantum efficiency (IQE). The simulation results suggest that the maximum is over 25 mA cm$^{-2}$ with optimized active layer thicknesses (Fig. 5b); whereas for the solar cell based on DC6-IC, the photocurrent can hardly go over 17 mA cm$^{-2}$.

We also derive IQEs for the solar cells by comparing the active layer absorption spectra simulated by the TM modeling with the experimentally determined EQEs. As shown in Fig. 5c, the lower $J_{sc}$ of solar cells based on DOC8-IC compared to that of the other NC-FREA-based solar cells, is related to a lower IQE. This agrees well with our previous PL quenching and R-SoXS measurements, suggesting a larger domain size and a less efficient exciton dissociation in the active layer based on DOC8-IC. The IQE of the solar cell based on DC6-IC is comparable to that of the DOC8-IC-based solar cell, but much lower than that of the other NC-FREA solar cells. This further limits the $J_{sc}$ of DC6-IC-based solar cells. In general, we can see that the IQEs of the NC-FREA-based solar cells are limited. The highest IQE of ~80%, leading to a high $J_{sc}$, is observed in the solar cell based on DOC2C6-2F, while IQEs of the solar cells based on other NC-FREAs are in the range of 60 to 70%.

**Voltage losses in the solar cells based on NC-FREAs.** Voltage losses are very low in the NC-FREA-based solar cells, especially for the solar cells based on DOC6-IC, DOC8-IC, and DOC2C6-IC. We realize $V_{oc}$ over 0.9 V in the solar cells with absorption onsets at around 900 nm. The voltage losses are analyzed by measuring the absorption and emission of the low-energy excitation state. We first measured electroluminescence (EL) of the devices with the neat and the blend active layers. From the EL spectra shown in Fig. 6a, we observe no difference between the emission of the blend and the neat material, indicating a negligible energy loss during the exciton dissociation/charge transfer

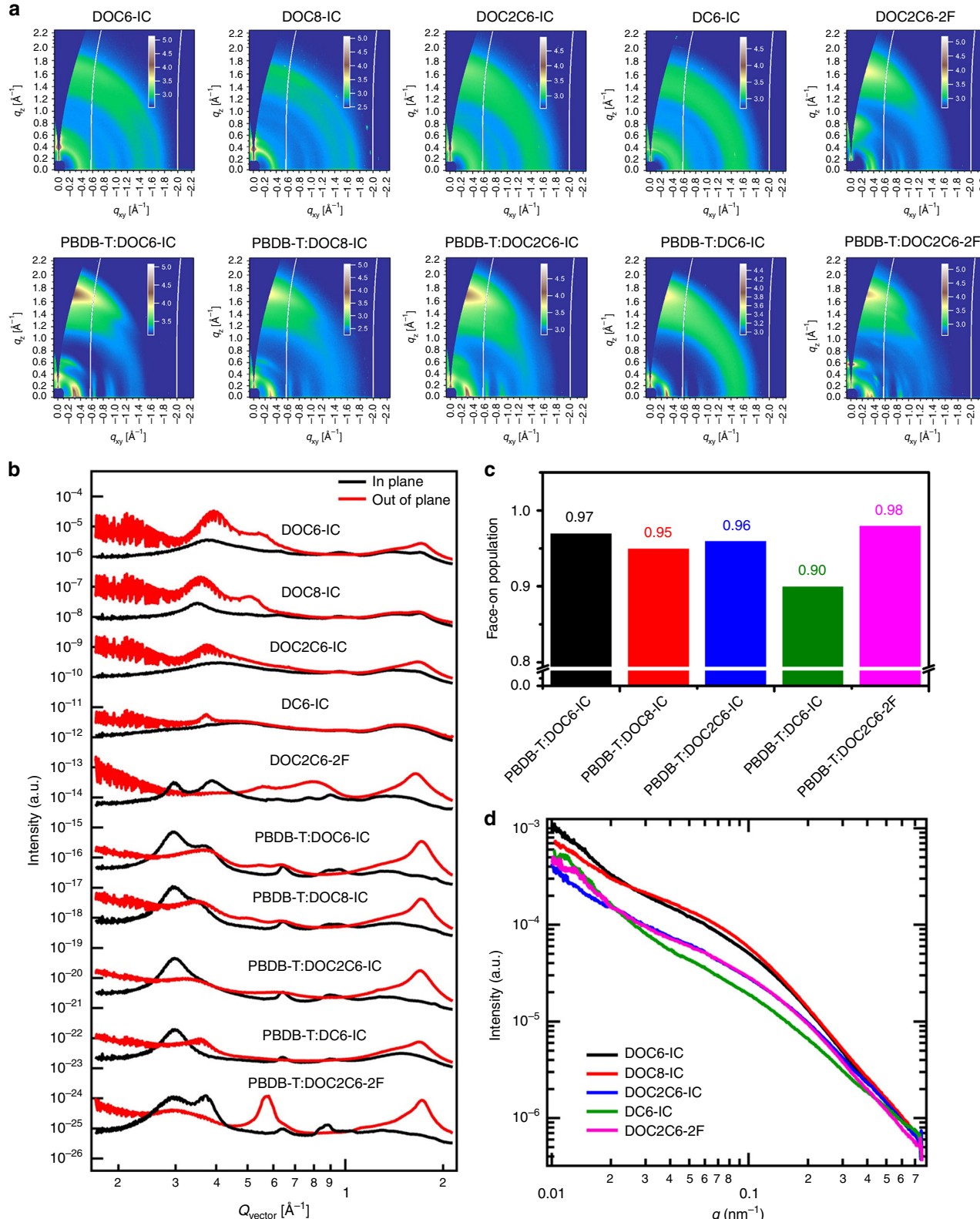

**Fig. 4** The morphology characterization of the active layers. **a** Two-dimensional GIWAXS patterns and **b** line profiles of neat and blend films of the small-molecule acceptors. **c** Calculated face-on population for PBDB-T:DOC6-IC, PBDB-T:DOC8-IC, PBDB-T:DOC2C6-IC, PBDB-T:DC6-IC, and PBDB-T:DOC2C6-2F blend films. **d** R-SoXS profiles in log scale for blend films

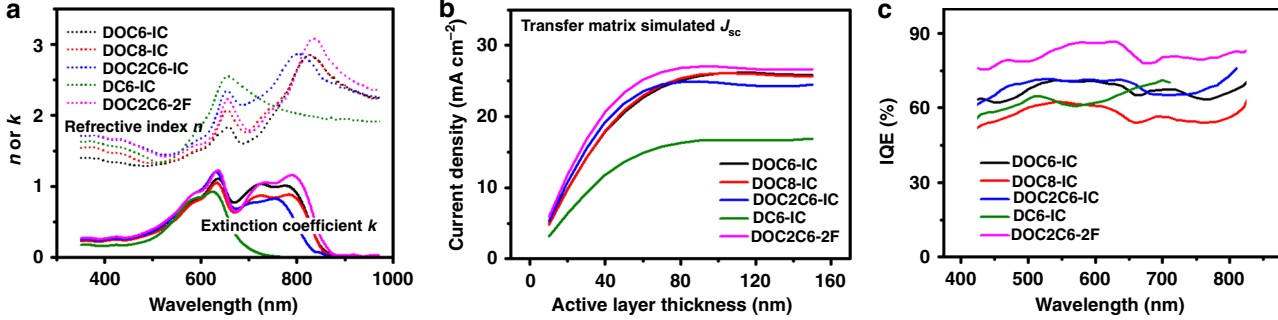

**Fig. 5** Optical properties of the materials and optical simulations. **a** Refractive indices and extinction coefficients of the blends based on NC-FREAs and DC6-IC. **b** Theoretical $J_{sc}$ as a function of the active layer thickness for the solar cells based on NC-FREAs and DC6-IC, simulated by a transfer matrix model. **c** IQE of the solar cells calculated using experimental EQE and simulated active layer absorption spectra. Source data are provided as a Source Data file

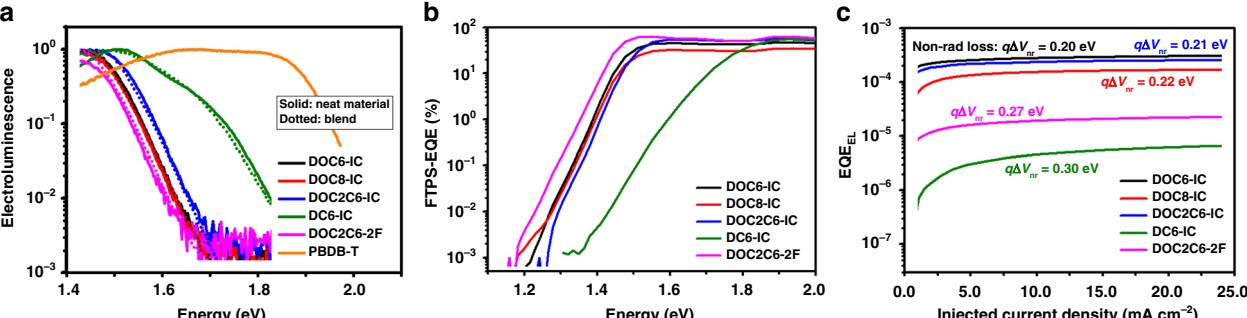

**Fig. 6** Voltage loss analysis of the NC-FREA solar cells. **a** Electroluminescence from solar cells based on different acceptors, recorded using an injection current of 1 mA. **b** EQE of the solar cells based on different acceptors at the absorption onset, measured using a sensitive FTPS method. **c** EQE$_{EL}$ of the solar cells based on different acceptors. Non-radiative recombination losses ($q\Delta V_{nr}$) calculated from EQE$_{EL}$ are also indicated. Source data are provided as a Source Data file

process[37]. Such a low energy loss is critically important for achieving high $V_{oc}$, and it is also confirmed by a sensitive Fourier transform photocurrent spectroscopy-EQE (FTPS-EQE) measurement[38]. As shown by the FTPS spectra in Fig. 6b, for all of the NC-FREA-based solar cells, we observe no sub-gap absorption feature that is commonly seen in blends with a large energetic offset between singlet and CT state excitons[39].

EL and FTPS-EQE spectra further suggest that the bandgap ($E_g$) is different in the solar cells based on different acceptors. The solar cell based on DC6-IC has the largest bandgap, with the absorption/emission onset blue shifted by over 0.2 eV, compared to that of the NC-FREA-based solar cells. Therefore, the solar cell based on DC6-IC is expected to have a significantly higher $V_{oc}$. However, compared to that of the NC-FREA solar cell based on DOC2C6-IC, $V_{oc}$ of the DC6-IC-based solar cell is only 0.05 V higher. This is because for the NC-FREA-based solar cells, recombination losses of photo generated charge carriers, limiting solar cell $V_{oc}$, are much lower than those in the solar cell based on DC6-IC.

$V_{oc}$ of a solar cell can be related to charge carrier recombination losses via

$$V_{oc} = V_{oc}^{rad} - \frac{kT}{q}\ln\left(\frac{1}{EQE_{EL}}\right) \quad (1)$$

where $k$ is the Boltzmann constant, $q$ is the elementary charge, and $T$ is temperature[39]. The first term on the right hand side of the equation, $V_{oc}^{rad}$, is the radiative recombination limit for $V_{oc}$, i.e., the maximum $V_{oc}$ achieved in the solar cell with no non-radiative recombination loss. The second term does not depend on $E_g$, but on the external quantum efficiency of electroluminescence (EQE$_{EL}$) of

the solar cell and it denotes the $V_{oc}$ loss induced by non-radiative recombination of charge carriers. It should be noted that EQE$_{EL}$ can be experimentally determined. As shown in Fig. 6c, EQE$_{EL}$ of the solar cells based on NC-FREAs and DC6-IC are plotted as a function of injection current density. Non-radiative recombination losses ($q\Delta V_{nr}$) are also indicated. We find that EQE$_{EL}$ of the NC-FREA solar cells based on DOC6-IC, DOC8-IC, and DOC2C6-IC, is significantly higher than that of DC6-IC-based solar cells, and non-radiative recombination losses are only around 0.2 eV in these NC-FREA solar cells, comparable to those in inorganic materials, such as CdTe or amorphous Si, based thin-film solar cells[40]. This is the main reason that for the very low voltage loss in these NC-FREA-based solar cells.

It should be noted that for the solar cell based on DOC2C6-2F, we observe lower EQE$_{EL}$, thus a higher non-radiative energy loss, compared to the other NC-FREA-based solar cells. This could be due to a small change of the energy levels, non-detectable in our spectroscopic measurements, induced by the modification of the end groups on the acceptor molecule. This could lead to a small energy difference between the singlet and the CT excitation state, thus reduce the probability of recombination via the more emissive singlet states. Although energy losses, limiting $V_{oc}$, are higher in the solar cell based on DOC2C6-2F, compared to those in the solar cells based on the other acceptors, the overall performance is higher with DOC2C6-2F due to higher IQE. This is not surprising, since a small energy offset between the singlet and the CT state has been shown to facilitate exciton dissociation and improve solar cell IQE[37,41].

**Transport and recombination of charge carriers in solar cells.** The hole and electron mobilities of the blend films are measured

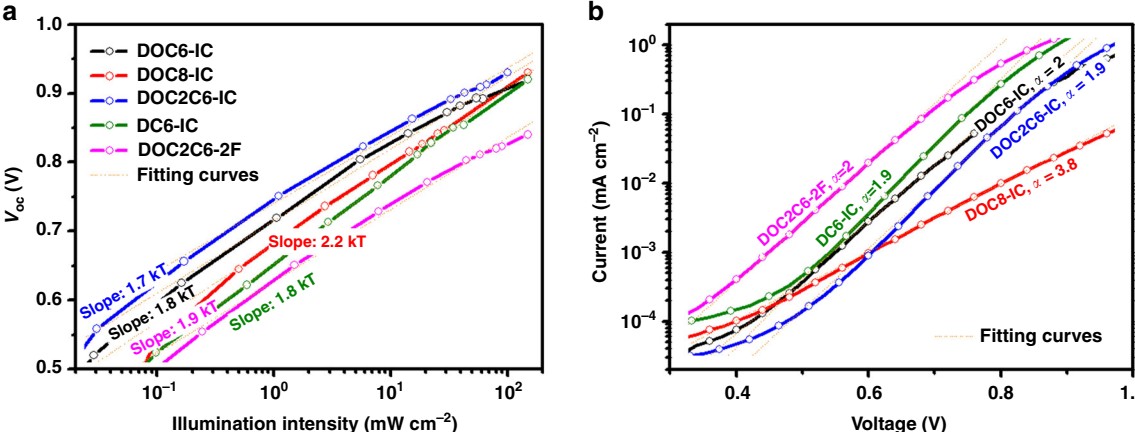

**Fig. 7** Charge carrier recombination in NC-FREA solar cells. **a** $V_{oc}$ as a function of illumination intensity for the solar cells based on different acceptors. Slopes are indicated. **b** Dark J–V characteristics of the solar cells based on different acceptors. Ideality factors, $\alpha$, are calculated using the equation: $J_{dark} = J_0 \exp(qV/\alpha kT)$, where $J_0$ represents the dark saturation current density

using single carrier solar cells and the space charge limited current (SCLC) method. The results are summarized in Supplementary Table 9 and Supplementary Fig. 10. Because of the poor planarity and conjugation of the molecular backbone, the unfused acceptor DC6-IC has the lowest electron mobility, compared to the NC-FREAs. Mobility of charge carriers of the non-fluorinated NC-FREAs is similar, on the order of $10^{-4}\ cm^2\ V^{-1}\ s^{-1}$. As for the fluorinated DOC2C6-2F, the mobilities are improved to the order of $10^{-3}\ cm^2\ V^{-1}\ s^{-1}$. Moreover, the mobilities in neat films are also measured by the SCLC method and summarized in Supplementary Table 10 and Supplementary Fig. 11.

To investigate the reason for the considerably lower FF in the solar cell based on DOC8-IC, we measure $V_{oc}$ of the solar cells as a function of illumination intensity. The results (Fig. 7a) suggest that charge carrier losses in the solar cell based on DOC8-IC are more dominated by a trap-assisted recombination mechanism, as indicated by the high slope of over $2.2kT$[42], whereas for the other solar cells, trap-assisted recombination losses are less significant. A similar conclusion is found in the exponential region of the dark J–V characteristics (Fig. 7b). An ideality factor of 3.8 is observed for the solar cell based on DOC8-IC, indicating a strong localization of charge carriers in this device, and the ideality factor is ~2 for the other solar cells[43].

## Discussion

In conclusion, we design and synthesize a series of NC-FREAs consisting of a central 2,5-bis(alkyloxy)phenylene unit, two flanking 4H-cyclopenta[2,1-b:3,4-b']dithiophene units, and two electron-withdrawing acceptor units. The S···O interactions between the central 2,5-bis(alkyloxy)phenylene unit and two flanking 4H-cyclopenta[2,1-b:3,4-b']dithiophene units produce a ladder-like structure that prevents the free rotation of the molecular backbone. Consequently, these NC-FREAs display broad absorption bands, high $\eta_{FL}$, and high electron mobilities in their blend films. The solubility of acceptors is finely tuned by changing the lateral substituents on the central phenylene unit. Therefore, the morphology of blend films can be readily optimized by changing the side chains on the central phenylene unit. We also synthesize DC6-IC with a central 2,5-dihexylphenylene unit and twisted molecular backbone as a control. DC6-IC shows a narrower absorption band, lower $\eta_{FL}$ yield, and lower electron mobility than those of the NC-FREAs. In comparison with DC6-IC-based solar cells, NC-FREA-based ones show relatively high $EQE_{EL}$ and very low non-radiative recombination voltage losses of only ~0.2 eV, which are comparable to those in inorganic

thin-film solar cells. As expected, solar cells containing the NC-FREAs achieve higher PCEs than that of the DC6-IC-based solar cell. The optimized solar cells with DOC2C6-IC and DOC2C6-2F display PCEs of 11.10% and 13.24%, respectively. To the best of our knowledge, a PCE of 13.24% is the highest value among all NC-FREAs. In addition, the synthesis of such NC-FREAs is simple compared with that of typical FREAs, which can drastically lower the cost of synthesis.

## Methods

**Fabrication of polymer solar cells.** Polymer solar cells were fabricated with the solar cell configuration of ITO/ZnO (30 nm)/active layer (70 nm)/MoO₃ (8.5 nm)/Ag (100 nm). ITO glasses with a conductivity of 15 Ω per square were cleaned according to the standard procedures before use. Precleaned ITO substrates were treated by UV-ozone for 20 min. A thin layer of ZnO was spin-coated on top of a cleaned ITO substrate at 3500 rpm for 40 s and annealed subsequently at 200 °C for 20 min on a hotplate before being transferred into a glove box. All the solutions (with 4 mg mL⁻¹ concentration) were heated at 110 °C for at least 30 min and then spin-coated on the ZnO layer. The top electrode was thermally evaporated, with an 8.5 nm MoO₃ layer, followed by a 100 nm Ag layer at a base pressure below $10^{-7}$ Torr. Six cells were fabricated on one substrate with an effective area of 0.04 cm². The area was measured by the optical microscope calibrated by a standard stage micrometer. The voltage scan started from −0.1 V to 1 V with a step of 0.02 V, adopting sweep scan mode with 1 ms of delay time between two measurement points. The measurement of the solar cells was conducted in a glove box without encapsulation. The temperature while measuring the J–V curves was ~25 °C.

**General measurements and characterization of devices.** UV–Visible absorption spectra were obtained on a PerkinElmer UV–Vis spectrometer model Lambda 750. Current–voltage characteristics were recorded under AM1.5 G AAA class solar simulator (SS-F53A, Enli Technology Ltd., Taiwan) with an intensity of 100 mW cm⁻² calibrated with a standard photovoltaic cell equipped with a KG5 filter (certificated by the National Institute of Metrology) and a Keithley 2400 source-measure unit. EQE data were obtained using a solar cell spectral response measurement system (QER3011, Enli Technology Co. Ltd). A calibrated silicon detector was used to determine the absolute photo sensitivity at different wavelengths. TEM images were obtained with a FEI Technai TF20 (Philip) transmission electron microscopy. AFM measurements were performed under ambient conditions using a Digital Instrument Multimode Nanoscope IIIA operating in the tapping mode. The thickness of the active layer was measured by a Dektak 6 M surface profilometer. The absolute fluorescence quantum yields were tested with Edinburgh FLS980 Spectrometer under the Xenon Xe source light path and NIRPMT-sphere detector light path. GIWAXS measurements were performed at beamline 7.3.3 at the Advanced Light Source. Samples were prepared on Si substrates using identical blend solutions as those used in devices. The 10 keV X-ray beam was incident at a grazing angle of 0.12° to 0.16°, selected to maximize the scattering intensity from the samples. The scattered X-rays were detected using a Dectris Pilatus 2M photon counting detector. R-SoXS transmission measurements were performed at beamline 11.0.1.2[2, 3] at the advanced light source (ALS). Samples for R-SoXS measurements were prepared on a poly(styrenesulfonate) (PSS) modified Si substrate under the same conditions as those used for device fabrication, and then transferred by floating in water to a 1.5 × 1.5 mm, 100 nm thick Si₃N₄ membrane supported by a 5 × 5 mm, 200 μm thick Si frame (Norcada

Inc.). Two-dimension scattering patterns were collected on an in-vacuum CCD camera (Princeton Instrument PI-MTE). The sample detector distance was calibrated from diffraction peaks of a triblock copolymer poly(isoprene-b-styrene-b-2-vinyl pyridine), which has a known spacing of 391 Å. The beam size at the sample is ~100 μm by 200 μm. ToF-SIMS analysis and depth profiling were conducted using a ToF-SIMS V spectrometer (IONTOF GmbH, Munster, Germany). Dual-beam experiments were performed using a 10 keV argon cluster ion beam ($Ar_n^+$) as sputtering beam and a 30.0 keV $Bi_3^+$ beam as analysis beam. High mass resolution spectra were collected in the positive ion mode over a $500 \times 500$ μm$^2$ area upon a pulsed analysis $Bi_3^+$ beam ($128 \times 128$ pixels per scan, pulse current = 1.1 to 1.2 pA, and cycle time = 100 μs) at the center of a $750 \times 750$ μm$^2$ crater eroded by $Ar_n^+$ sputtering source. The current of the $Ar_n^+$ was ~5.6 nA with lead-off time 18 μs. The mass resolution ($m/\Delta m$) at $m/z$ 29.04 ($C_2H_5^+$) was 5000 to 6000. Charge compensation was always applied during the experiments to prevent charge accumulation. Each sample was collected three times at different areas. Positive ion spectra were recorded and calibrated by $CH_3^+$, $C_2H_5^+$, and $C_3H_5^+$. Signals were collected layer by layer, and the profiles of ion intensity versus sputter time were generated by ION-TOF SurfaceLab (Version 7.0). The contact angle were measured by OCA20 instrument (Dataphysics, Germany).

To determine optical constants of the materials used in this work, three thin-films with different thicknesses were prepared on clean glass substrates for each blend system. Reflectance and transmittance spectra of the thin-films were measured, and used in a transfer matrix model to calculate the $n$ and $k$ constants. Electroluminescence spectra were measured using a Kymera-328I spectrograph and an EMCCD (DU970P-BVF) camera from Andor Technology. A constant injection current of 1 mA was used, for all of the samples. $EQE_{EL}$ was measured using a home built set-up. A Keithley 2400 was used to drive the solar cells studied in this work, and a picoammeter Keithly 6482 was used to record photocurrent from a Si detector (HAMAMATSU S1337-1010BQ). Illumination intensity-dependent $V_{oc}$ measurements were done using a halogen lamp, and a set of neutral density filters, and device $V_{oc}$ were recorded by a Keithley 2450. FTPS were taken using a FTIR spectrometer (Nicolets is 50) with an external Si detector, HAMAMATSU S1337-1010BQ. The photocurrent response was amplified by a Stanford SR570 pre-amplifier.

**Synthesis of the small-molecule acceptors**. The detailed synthesis routes can be found in the Supplementary Methods. The nuclear magnetic resonance data, matrix-assisted laser desorption/ionization time-of-flight mass spectrometry (MALDI-TOF MS) data are included in the Supplementary Methods.

## Data availability

The source data underlying Figs. 2a, b, 3a, b, 5a–c, and 6a–c, Table 1, and Supplementary Figs. 2, 3, 4a, b, 7a, b, 9, 10, and 11 are provided as a Source Data file.

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

## Acknowledgements

Financial support from the National Natural Science Foundation of China (21734009, 21421003, 91233205, 20774099, 21574013, 51673028, and 21875182) is gratefully acknowledged. We thank Dr. Chenhui Zhu for GIWAXS measurement. X-ray data was acquired at beamlines 7.3.3 and 11.0.1.2 at the Advanced Light Source, which is supported by the Director, Office of Science, Office of Basic Energy Sciences, of the U.S. Department of Energy under Contract No. DE-AC02-05CH11231. We thank Chenhui Zhu at beamline 7.3.3, and Cheng Wang at beamline 11.0.1.2 for assistance with data acquisition. We also thank Dr. Yao Zhao from the Institute of Chemistry, Chinese Academy of Sciences, for the ToF-SIMS analysis.

## Author contributions

H.H. designed and synthesized the small-molecule acceptors. Q.G. and C.Z. fabricated and characterized the devices and carried out the detailed characterizations. S.F. synthesized the polymer donor. Z.B. and W.X. carried out the R-SoXS and GIWAXS measurements. J.Y. calculated the optical constants of the blend materials, performed electroluminescence, $EQE_{EL}$, FTPS-EQE measurements, and determined ideality factor from illumination intensity dependent and dark $J$–$V$ measurements. J.S. carried out the density functional theory (DFT) calculations. H.H. and Q.G. prepared the manuscript. C. L. directed the synthesis of the project. W.M. directed the R-SoXS and GIWAXS measurements of the project. Z.T. performed optical transfer matrix simulations and calculated IQE. X.X. and Z.B. supervised and directed this project. All authors discussed the results and commented on the manuscript.

## Additional information

**Competing interests:** The authors declare no competing interests.

