## [Peer Review File · Nature Communications]

Reviewers' comments:

Reviewer #1 (Remarks to the Author):

For the most part this is a solid paper that describes some new non-fullerene acceptors that exhibit very nice performance in organic solar cells. However, I don't think that the paper is really at the level that Nature Communication is the appropriate venue for it. The field of non-fullerene acceptors has exploded in recent years and the number of report of materials with efficiencies surpassing 12 percent efficiency itself is quite significant. Therefore publication in Nature Communication really requires in my view a significant conceptual advance over what has been done before. In my view, in this paper one would need to argue that the conceptual advance was the ability to get a highly planar structure using alkoxy-thiophene interactions. Had this been the first report of this design strategy for obtaining rigid organic molecules, publication in Nature Communication would likely have been warranted. However as the authors themselves note, such an approach was introduced to the organic electronics community quite some time ago, and even by the authors themselves. Thus the authors are essentially applying a well-known structural control mechanism to make another example of a rigid non-fullerene acceptor, which is as they note somewhat easier to synthesize, (although they overstate the complexity of synthesizing the FREAs, from my experience).

For these reasons, I think that this work is more suitable for publication in Chemistry of Materials, Advanced Functional Materials or Journal of Materials Chemistry A.

The authors should also take note that several of the compounds reported in this paper are new and as such prior to publication in most any journal should be more completely characterized. In particular while some compounds are not "final" products they authors should collect and report carbon NMR spectra and mass spectrometry data as well. Note that in the supplementary materials the authors refer to ^{12}C NMR, which is clearly in error. First it is ^{13}C , and second the spectra are undoubtedly proton decoupled and as such they should be reported as $^{13}\text{C}\{^1\text{H}\}$. Also in my view the data are sufficiently good that that the authors could do a more complete job in reported coupling constants in for the proton NMR spectra.

Reviewer #2 (Remarks to the Author):

Developing low-cost electron acceptors is a challenging task in the field of organic solar cells (OSCs). In the manuscript, Hao Huang et al. tried to address the above problem and developed a series of electron acceptors featuring a central "supramolecular ladder-type core". In my opinion, the design strategy used in this work is more novel compared with the numerous reports on modified fused-ring electron acceptors. Moreover, the authors clearly showed that the new design is quite successful and creates the highly efficient acceptor DOC2C6-2F. Most interestingly, these acceptors show high electron mobility on the order of 10^{-3} ($\text{cm}^2\text{V}^{-1}\text{s}^{-1}$). It is also claimed that these devices show very low non-radiative recombination voltage loss of ~ 0.2 eV, comparable to those in inorganic solar cells. Overall, this work is generally performed and described well and is likely to be of high interest to the readers of Nature Communications. I conditionally recommend it for publication.

There are several important issues that should be fully addressed.

Nomenclature:

* The terms "supramolecular" and "ladder-type core" are completely misleading if not dead wrong. What the authors have is a single molecule, constructed with covalent bonds. These molecules exhibit

molecular packing, i.e. exhibit supramolecular self-assembly, just like any other acceptor. If that supramolecular self-assembly is enhanced by molecule design, it does not make it a supramolecular acceptor. It might exhibit D-A enhance aggregation. The nomenclature chosen should clearly not violate the definition of “supramolecular”.

* similar “ladder-type” implies that it is a structure that is similar to a ladder. However, the core has two “ladder-elements” that are connected by units and bonds that would allow twisting. I understand that the authors try to say, but a better term needs to be invented if they want to “brand” their molecular motif.

Several key technical and contextual points that need to be addressed:

- The authors emphasized low-cost of their supramolecular acceptors in several places, e.g., “The use of supramolecular interactions avoids the tedious synthesis of covalently fused ring structures and markedly lowers the synthetic cost”, “However, the synthesis of such large fused-ring structures is difficult and expensive”, “drastically lower the cost of synthesis”, “Importantly, the synthesis of such supramolecular acceptors is easier and cheaper than that of covalent acceptors”. A quantitative analysis of the cost of supramolecular acceptors and fused-ring acceptors is critical and thus needed.
- In page 8, the authors claimed that “supramolecular acceptor based ones show relatively high EQEEL and very low non-radiative recombination voltage losses of only ~ 0.2 eV, which are comparable to those in inorganic thin-film solar cells.” The authors should explicitly point out the references on inorganic solar cells. Moreover, and very importantly, the EQEEL has a current density dependence. It should be made explicit what the comparable current density in a OPV devices under 1.5AM radiation would be. Furthermore, and even more importantly, the authors just focus on one loss term. They also need to consider the loss due to gap states (DOI: 10.1002/aenm.201501721). Clearly, these materials have substantial energetic disorder. This disorder should be quantified by fitting the tails in the bandgap and reporting an Urbach energy. Finally, it should be discussed if the high EQEEL might be coupled inversely to the energetic disorder and if there is thus a competing design conundrum (low disorder, high EQE), of which they have solved only one aspect. I think such an analysis and discussion would add strength to the manuscript.
- the author use a numerical value for the bandgap, when discussing losses. However, given the strong exponential tail, there is no obvious bandgap. The rules by which bandgap has been evaluated needs to be articulated and put in the context of the literature.
- In Page 5, “special orientation” and “enhanced face-on orientation” are vague. I would suggest the authors provide a precise definition of those terms and avoid confusion. Quantifying the face-on to edge-on ratio is highly recommended for all samples. A comparison with literature results would strengthen the analysis (Vohra et al. Nature Photonics, 2015, 9, 403-408; etc).
- The authors concluded that the special orientation of DOC2C6-2F blend film benefits the three-dimensional transport of charge in the device, thus leading to an excellent FF, while there is no justification for this. To my knowledge, the benefits of mixed face-on/edge-on in organic solar cells were demonstrated by Yang et al. Energy Environ. Sci., 2017, 10, 258-265 and other papers. In particular, prior studies (J. Mater. Chem. A, 2016, 4, 16335–16340; J. Am. Chem. Soc., 2017, 139, 5085–5094) suggested that three-dimensional charge pathways provided by this coexistence texture (comprising face-on and edge-on crystallites) are beneficial for nonfullerene solar cells.
- In terms of practical application, the operation stability of these devices needs to be evaluated and reported.
- In their analysis, authors mainly attributed the performance difference to molecular packing and established the qualitative correlation. Other parameters can be very significant. Hence, vertical composition distribution and miscibility/domain purity (J. Am. Chem. Soc. 2012, 134, 15869-15879; Nature Materials, 2018, 17, 253-260) information should be evaluated and provided.
- As the molecular weight of PBDB-T type conjugated polymers likely affects the device reproducibility and many characterizations (e.g., DOI: 10.1016/j.joule.2018.11.006; DOI: 10.1002/solr.201800129). The authors should add and discuss such information.

Reviewer #3 (Remarks to the Author):

Most non-fullerene acceptors developed to date for high-performance organic solar cells feature a fused ladder-type electron-rich core. In this work, Bo and co-workers developed a series of non-fullerene acceptor without such fused core. The incorporation of alkoxy chain triggers intramolecular non-covalent S-O interaction, locking the acceptor materials backbone to achieve planar conformation. In addition, this dynamic non-covalent interaction can greatly reduce the synthetic complexity and hence the materials can be accessed at low-cost. The strong electron-donating ability of alkoxy chain markedly broadens the absorption to the near-infrared region and the planer backbone facilitates charge transport. When incorporated into photovoltaic devices, the optimized solar cells with PBDB-T:DOC2C6-2F active layer attained a remarkable power conversion efficiency (PCE) of >13%. Although a PCE of >15% has been achieved in non-fullerene solar cells very recently, this device performance is still highly promising in terms of the chemical structure novelty and materials design approach. Moreover, various materials and device characterization techniques were well performed, which help establish the structure-property correlations and offer insights into future materials design. In principle, the manuscript is well-written and the data are supportive to the conclusion. Hence, publication of this manuscript is recommended in this journal after the following issues are addressed.

- 1) I am not sure if this series of materials can be called supramolecular electron acceptors since the non-covalent S-O interaction is intramolecular rather than intermolecular (or supramolecular).
- 2) The study showed a low non-radiative recombination of energy loss in the range of 0.20 to 0.27 eV, the overall energy loss should be calculated and included.
- 3) Please provide the more detailed information for intermediates, such as ¹³C NMR, element analyses and HRMS. Scheme 1. in vi) a) chloroform, reflux; b) chloroform, r.t., this is confusing. The structure of polymer donor should be included somewhere in the main text.
- 4) Figure 1b shows the absorbance coefficient of solution, while the absorption of film is normalized. The film absorption should also be based on absorbance coefficient since solar cell is film-based device. In addition, I would like to see UV-Vis of the blends-how does the absorption strength of these "supramolecular electron acceptors" compared to that of the PBDB-T?
- 5) The non-covalent S-O interaction (or conformation lock) has been known for a long time, the first usage of this approach for developing high-performance organic semiconductors was pioneered by Prof. Mark D. Watson, which should be mentioned in the manuscript. In addition, two highly relevant works, J. Am. Chem. Soc. 2013, 135, 1986. and Adv. Mater. 2016, 28, 9969. which should be cited in appropriate places.
- 6) On Page #1 "....., we for the first time developed a kind of supramolecular, the intramolecular noncovalent interactions in which locked the aromatic units in the molecular backbone to form supramolecular fused-ring structures." Please re-phrase this sentence. On same page, "....two branched octyl side chains...." I don't think that you can call 2-ethylhexyl as a branched octyl chain.
- 7) ".....display broad structured absorption ranging from 500 to 900 nm with two peaks....". One should be absorption shoulder.
- 8) The J_{sc} values from J-V curves should be included in Table 1.
- 9) From DOC6-IC to DOC8-IC and to DOC2C6-IC, the chain becomes more bulkier, however the device performance didn't show a monotonic increasing trend, please comment on this. Particularly, the image between film morphology and device performance is still not very clear. The RoSX can provide information on domain size and purity, can author include these data?
- 10) The authors claimed that the all blend films are highly smooth, the specific data should be included.

- 11) "...Meanwhile the outstanding (010) peak in the OOP direction shows an enhanced face-on orientation of DOC2C6-2F." The driving force for such microstructural feature?
- 12) More background for transfer matrix simulation should be provided, particularly how the maximum photocurrent was derived.
- 13) There are a new series of electron acceptors, the mobilities in neat films should be also measured by the SCLC method.

Reviewer #4 (Remarks to the Author):

This manuscript reports a series of "supramolecular electron acceptors" (SMEAs) containing alkoxy side chains and dicyanoindanone end groups are synthesized, characterized, and used for the fabrication of solar cells. Compared to the more commonly employed fused-ring electron acceptors (FREAs), the authors claim that the SMEAs are easy and cheap to synthesize, by avoiding the need to build a system of fused rings. In addition, the introduction of the alkoxy side in the donor core is postulated to lead to an S-O conformation locking interaction, reinforcing a planar conformation that is crucial for π -electron delocalization. Here, a variety of techniques to study the neat material and blends with the polymer PBDB-T, including absorption spectroscopy, GIWAXS, AFM, TEM, photophysical measurements, and SCLC mobility to study their molecules and system. Through their characterization, the authors found that the SMEA DOC2C6-2F exhibited the highest PCE of the molecules under study, although the reason for this is not the focus of the manuscript. Instead, the authors choose to focus on how the SMEAs differ from a control "non-SMEA" (referred to as DC6-IC) that is nearly identical to the SMEAs, except that the core alkoxy side chains have been replaced with a hexyl side chain. The authors find that compared to the SMEAs, DC6-IC has a blue shifted, more narrow absorption profile, greater non-radiative recombination losses, and lower FF and J_{sc} .

This work could be suitable for publication after addressing the points below.

Specific Comments

- 1) The authors make the claim several times throughout the manuscript that synthesis of SMEAs is simple and cheap compared to that of FREAs. However, no evidence or cost analysis of this is provided. In fact, compared to ITIC-4F, a well-known acceptor that performs comparably to the best molecule presented here, the synthetic sequence and cost appear similar at best. To make the core of ITIC-4F, thienothiophene is coupled to 2,5-dibromoterephthalic acid diethylester (which is commercially available and can be derived from the corresponding acid, which is available for <\$30/10 g) by Negishi coupling. The alkyl chains are then added in, and the compound is cyclized with acid in 1 pot. The remainder of the syntheses are identical: formylation followed by Knoevenagel condensation of the same difluorinated dicyanoindanone. This is a total of 4 steps, or 5 if the addition and cyclization are performed separately. In the synthesis outlined for the SMEAs, 4 steps are required. In addition, the cyclopentadithiophene compound used for the Stille coupling (which itself is extremely costly due to the use of toxic tin compounds and should be replaced with a Negishi reaction) is ~4x as expensive as thienothiophene (\$400/g vs \$103/g from sigma aldrich) which is used for the synthesis of most FREAs. This difference in cost reflects the long sequence required for the synthesis of cyclopentadithiophene compounds relative to thienothiophene. If the authors wish to keep the claim that SMEAs are cheap and easy to make, they need to consider not only the step count from commercially available components, but the cost and synthetic challenges associated with the components they are using.
- 2) Although not clearly stated, the authors make a distinction between DC6-IC (non-SMEA) and the all the other molecules (SMEAs) in the manuscript on the basis that DC6-IC does not contain S-O interactions that lock the planarity of the donor core into place. This reviewer takes issue with this

distinction. All molecule reported in this manuscript almost certainly contain S-O conformation locking interactions between the dicyanoindanone end group and the cyclopentadithiophene core group. The chemical structures in scheme 1 should be redrawn to reflect this. Furthermore, conformation locking interaction have been reported in numerous polymers and small molecule semiconductors (review: DOI: 10.1021/acs.chemrev.7b00084). More to the point, S-O conformation interactions have been directly observed in one subgroup of FREA crystal structures (DOI:

<https://doi.org/10.1073/pnas.1807535115>), and is undoubtedly present in many others. Should these molecules also be considered SMEAs because of the conformation locking non-covalent interactions?

3) In the fluorescence quenching experiments, the exact percentage of fluorescence quenching should be determined.

4) Based on the fluorescence quenching experiments, the authors propose that DOC8-IC forms much larger domains than the others, as the quenching of the DOC8-IC by PBDB-T is much less than the others. This is plausible, but the TEM images seem to show no difference and the CLs measured by GIWAXS are fairly similar between DOC8-IC and the others. RSoXS or another technique should be performed to clearly measure the domain size distribution in the blends.

5) The GIWAXS images are small and of low resolution. They are hard to read, and when zooming in the images become blurry.

6) The authors state that the molecules DOC8-IC and DC6-IC exhibit no preferred orientation on the basis of the GIWAXS data. Does this mean isotropic distribution?

7) FTPS-EQE should be defined when first used.

Reviewers' comments:

Reviewer #1 (Remarks to the Author):

For the most part this is a solid paper that describes some new non-fullerene acceptors that exhibit very nice performance in organic solar cells. However, I don't think that the paper is really at the level that Nature Communication is the appropriate venue for it. The field of non-fullerene acceptors has exploded in recent years and the number of report of materials with efficiencies surpassing 12 percent efficiency itself is quite significant. Therefore publication in Nature Communication really requires in my view a significant conceptual advance over what has been done before. In my view, in this paper one would need to argue that the conceptual advance was the ability to get a highly planar structure using alkoxy-thiophene interactions. Had this been the first report of this design strategy for obtaining rigid organic molecules, publication in Nature Communication would likely have been warranted. However as the authors themselves note, such an approach was introduced to the organic electronics community quite some time ago, and even by the authors themselves. Thus the authors are essentially applying a well-known structural control mechanism to make another example of a rigid non-fullerene acceptor, which is as they note somewhat easier to synthesize, (although they over-state the complexity of synthesizing the FREAs, from my experience). For these reasons, I think that this work is more suitable for publication in Chemistry of Materials, Advanced Functional Materials or Journal of Materials Chemistry A.

Answer:

Thanks for the comment. We first proposed the use of intramolecular secondary

interactions to assemble noncovalently fused-ring electron acceptors (NC FREAs) in our previous article (*J. Am. Chem. Soc.* 2017, 139, 3356) and we are glad that this concept is well noticed and developed. However, the study of such acceptors is now stuck in unsatisfactory PCE and confused device physics. Here, we designed a series of new high-efficiency NC FREAs which have never been reported and aimed to further study the effect of noncovalent interactions. Compared with our previously reported NC FREAs (*J. Am. Chem. Soc.* 2017, 139, 3356), the design strategy introduced here for constructing NC FREAs only employs a small aromatic ring instead of large fused aromatic rings as the D unit and then utilizes intramolecular noncovalent (secondary) interactions to lock the planar conformation of the central π -D- π unit in the A- π -D- π -A type molecules. For the first time, a near-infrared absorption and a high PCE over 13% are achieved in our NC FREAs. It should be stressed that in this article we confirmed the positive effect of the introduction of intramolecular noncovalent interaction in both aspect of chemistry and device physics. With these NC FREAs designed in purpose, we discussed the device performance in detail and depth, and we show extremely low voltage losses in the solar cells based on NC FREAs. Thus, we believe the work will raise people's attention to NC FREAs for many advantages mentioned in our manuscript.

The authors should also take note that several of the compounds reported in this paper are new and as such prior to publication in most any journal should be more completely characterized. In particular while some compounds are not "final" products they authors should collect and report carbon NMR spectra and mass spectrometry data as well. Note

that in the supplementary materials the authors refer to ^1C NMR, which is clearly in error. First it is ^{13}C , and second the spectra are undoubtedly proton decoupled and as such they should be reported as $^{13}\text{C}\{^1\text{H}\}$. Also in my view the data are sufficiently good that that the authors could do a more complete job in reported coupling constants in for the proton NMR spectra.

Answer:

Thanks for the reviewer's nice advice. We have provided the ^{13}C NMR spectra and MALDI-TOF data in the revised supporting information section "1.7 synthesis" (Page S5-S14). And we also corrected the mistake of " ^1C NMR" in the supporting information section "1.7 synthesis" (Page S5-S14). The coupling constants were also added in the section "1.7 synthesis" (Page S5-S14).

Reviewer #2 (Remarks to the Author):

Developing low-cost electron acceptors is a challenging task in the field of organic solar cells (OSCs). In the manuscript, Hao Huang et al. tried to address the above problem and developed a series of electron acceptors featuring a central "supramolecular ladder-type core". In my opinion, the design strategy used in this work is more novel compared with the numerous reports on modified fused-ring electron acceptors. Moreover, the authors clearly showed that the new design is quite successful and creates the highly efficient acceptor DOC2C6-2F. Most interestingly, these acceptors show high electron mobility on

the order of 10^{-3} (cm²V⁻¹s⁻¹). It is also claimed that these devices show very low non-radiative recombination voltage loss of ~ 0.2 eV, comparable to those in inorganic solar cells. Overall, this work is generally performed and described well and is likely to be of high interest to the readers of Nature Communications. I conditionally recommend it for publication.

There are several important issues that should be fully addressed.

Nomenclature:

** The terms "supramolecular" and "ladder-type core" are completely misleading if not dead wrong. What the authors have is a single molecule, constructed with covalent bonds. These molecules exhibit molecular packing, i.e. exhibit supramolecular self-assembly, just like any other acceptor. If that supramolecular self-assembly is enhanced by molecule design, it does not make it a supramolecular acceptor. It might exhibit D-A enhance aggregation. The nomenclature chosen should clearly not violate the definition of "supramolecular".*

** similar "ladder-type" implies that it is a structure that is similar to a ladder. However, the core has two "ladder-elements" that are connected by units and bonds that would allow twisting. I understand that the authors try to say, but a better term needs to be invented if they want to "brand" their molecular motif.*

Answer:

Thanks for the reviewer's valuable comment. We have changed the title of this manuscript to "Noncovalently Fused-Ring Electron Acceptors with Near-Infrared Absorption for High-Performance Organic Solar Cells with Low Voltage Losses". And "ladder-type" is changed to "ladder-like" (E. W. Meijer et al. *J. Am. Chem. Soc.* **1996**, *118*, 8717-8718. π -Conjugated Oligomers and Polymers with a Self-Assembled Ladder-like Structure; *Chem. Eur. J.* **2000**, *6*, 4597-4603. Ladder-like oligomers; Intramolecular hydrogen bonding, push-pull character, and electron affinity). We have rephrased the descriptions in the revised manuscript.

The updated sentences are as follows:

"a central ladder-like core and two dicyanoindanone electron-withdrawing groups. As shown in Scheme 1, the ladder-like core is composed of a central ... a planar ladder-like structure".

Several key technical and contextual points that need to be addressed:

- The authors emphasized low-cost of their supramolecular acceptors in several places, e.g., "The use of supramolecular interactions avoids the tedious synthesis of covalently fused-ring structures and markedly lowers the synthetic cost", "However, the synthesis of such large fused-ring structures is difficult and expensive", "drastically lower the cost of synthesis", "Importantly, the synthesis of such supramolecular acceptors is easier and cheaper than that of covalent acceptors". A quantitative analysis of the cost of supramolecular acceptors and fused-ring acceptors is critical and thus needed.

Answer:

Thanks for the nice suggestions. To illustrate that the synthesis of noncovalently fused-ring acceptor (**A1**) is cheaper than that of the corresponding fused-ring acceptor (**A2**), we outlined their synthetic routes in Scheme A. Starting from compound **a**, it requires three steps to achieve the aimed acceptor **A1** as shown in route a. Each step can be accomplished in relatively high yield. As for **A2**, two extra steps are required to obtain covalently fused structure as shown in route b. According to the literature, the total yield of these two extra steps is usually low (ca. 30% to 50%).

Scheme A

- In page 8, the authors claimed that “supramolecular acceptor based ones show relatively high EQEEL and very low non-radiative recombination voltage losses of only ~0.2 eV, which are comparable to those in inorganic thin-film solar cells.” The authors should explicitly point out the references on inorganic solar cells.

Answer:

Thanks for the reviewer's nice suggestion. We have pointed out the reference on inorganic solar cells (Page 8). The revised sentences are as follows:

"... non-radiative recombination losses are only ~ 0.2 eV in these NC FREA solar cells, comparable to those in inorganic materials, such as CdTe or amorphous Si based thin-film solar cells.⁴⁰"

Moreover, and very importantly, the EQEEL has a current density dependence. It should be made explicit what the comparable current density in a OPV devices under 1.5AM radiation would be.

Answer:

Thanks for the reviewer's suggestion. The active area is 0.04 cm^2 for devices studied in this work. Thus, the device photocurrent is about 0.5 mA under 1.5 AM radiation, which is at the center of the x-axis of the EQEEL plot. We have changed Figure 5c with the abscissa to current density. (Page 7)

Furthermore, and even more importantly, the authors just focus on one loss term. They also need to consider the loss due to gap states (DOI: 10.1002/aenm.201501721). Clearly, these materials have substantial energetic disorder. This disorder should be quantified by fitting the tails in the bandgap and reporting an Urbach energy. Finally, it should be discussed if the high EQEEL might be coupled inversely to the energetic disorder and if there is thus a competing design conundrum (low disorder, high EQE), of which they have

solved only one aspect. I think such an analysis and discussion would add strength to the manuscript.

Answer:

Thanks for the reviewer's nice suggestion. Our analysis on voltage losses is based on the detailed balance theory, (Phys. Rev. B 81, 125204), that assumes charge carriers recombine via CT states. Thus CT state emission efficiency determines device EQE-EL and thus the non-radiative voltage losses. Meanwhile, although it is not discussed in this manuscript, the radiative recombination voltage losses are also determined by the CT state property, i.e. radiative decay rate of CT excitation. (DOI: 10.1002/adma.201400114)

However, it should be noted that it is still not very clear regarding the reason for the generally lower EQEEL of BHJ organic solar cells, compared to that of inorganic solar cells. K. Vandewal proposed that non radiative recombination can be assisted by the vibration modes of the organic molecules (DOI: 10.1038/nenergy.2017.53); D. Qian showed that hybridization of CT and singlet excitation states can lead to reduced non-radiative recombination losses (DOI: 10.1038/s41563-018-0128-z); and as pointed out by the reviewer, disorder could also be related to the non-radiative voltage losses, although it is not considered in the theory that we used for analyzing voltage losses. It is extremely challenging to fully explain the reason for the high EQEEL of the NC FREA based solar cells. Even if it is only to give a suggestion, it would require a substantial amount of theoretical and experimental work, which would, in our opinion, be better suited for a separate publication.

Nevertheless, we now report the value of Urbach energy for the **DOC6-IC**, **DOC8-IC**,

DOC2C6-IC, DC6-IC and DOC2C6-2F, which are 0.027, 0.025, 0.036, 0.072, and 0.026 eV, respectively. See supporting information Table S4.

- the author use a numerical value for the bandgap, when discussion losses. However, given the strong exponential tail, there is no obvious bandgap. The rules by which bandgap has been evaluated needs to be articulated and put in the context of the literature.

Answer:

Thanks for the reviewer's suggestion. We should note that although we gave numerical values for the bandgaps, they were not needed for analyzing voltage losses. We agree with the reviewer that the value for the bandgap is difficult to define, and thus we no longer show the numerical values for the bandgaps in the main text of the revised manuscript.

(See SI)

- In Page 5, "special orientation" and "enhanced face-on orientation" are vague. I would suggest the authors provide a precise definition of those terms and avoid confusion. Quantifying the face-on to edge-on ratio is highly recommended for all samples. A comparison with literature results would strengthen the analysis (Vohra et al. Nature Photonics, 2015, 9, 403-408; etc).

Answer:

Thanks for the reviewer's suggestion. The "special orientation" means "an ordered packing along the molecular backbone direction" in the DOC2C6-2F film described

several sentences ahead. To avoid confusion, we have changed the words “special orientation” to “special orientation (an ordered packing along the molecular backbone direction)”. Also, thanks the advice for the face-on to edge-on ratio. We added such data in the revised manuscript to confirm our opinion in the Figure 3c and discussed in Page 5 and 6.

We add this information that “The face-on population (Figure 3) was calculated from $A2/(A1+A2)$ to be 0.97, 0.95, 0.96, 0.90, and 0.98 for DOC6-IC, DOC8-IC, DOC2C6-IC, DC6-IC and DOC2C6-2F, respectively.^{30,31} “A1 is the area of 45°-90° which represents the edge-on region, A2 is the area of 0°- 45° which represents the face-on region.”

- The authors concluded that the special orientation of DOC2C6-2F blend film benefits the three-dimensional transport of charge in the device, thus leading to an excellent FF, while there is no justification for this. To my knowledge, the benefits of mixed face-on/edge-on in organic solar cells were demonstrated by Yang et al. Energy Environ. Sci., 2017,10, 258-265 and other papers. In particular, prior studies (J. Mater. Chem. A, 2016, 4, 16335–16340; J. Am. Chem. Soc., 2017, 139, 5085–5094) suggested that three-dimensional charge pathways provided by this coexistence texture (comprising face-on and edge-on crystallites) are beneficial for nonfullerene solar cells.

Answer:

Thanks for the reviewer's nice advice. We want also to express the same viewpoint and have cited the references mentioned by the reviewer (Ref. 32-34).

- In terms of practical application, the operation stability of these devices needs to be evaluated and reported.

Answer:

Thank for the nice advice. We are sorry that currently we are unable to do this test at present time due to the lack of related instruments. We hope we can complete the test in the near future.

- In their analysis, authors mainly attributed the performance difference to molecular packing and established the qualitative correlation. Other parameters can be very significant. Hence, vertical composition distribution and miscibility/domain purity (J. Am. Chem. Soc. 2012, 134, 15869-15879; Nature Materials, 2018, 17, 253-260) information should be evaluated and provided.

Answer:

Thanks for the nice advices.

1. We adopted the time of flight secondary ion mass spectroscopy (ToF-SIMS) to study the vertical composition distribution and find the acceptor and donor uniformly distributed in the vertical direction. We have added this information in the revised manuscript (Page 6) and supporting information in Figure S9.

Details are as follows: "We used the time of flight secondary ion mass spectroscopy (ToF-SIMS) to study the vertical composition distribution in the active layer. The vertical composition distribution could be determined by the intensity of CN⁻ signal, since the signal of CN⁻ is only from the small molecular acceptors (the intensity of CN⁻ for neat

PBDB-T film is markedly lower than the active layers). The intensity of CN⁻ in different depth of active layers shows no obvious change, indicating that the acceptor and donor uniformly distributed in the vertical direction (see Figure S9).”

2. The domain size was investigated by R-SoXS and the corresponding discussions are added in Page 4 and Figure 3d. Details are as follows:” We find that the active layer of DOC8-IC shows larger domain size of 25.7 nm, while those of DOC6-IC and DOC2C6-IC exhibit smaller domain size of 22.7 nm and 15.7 nm, respectively. Due to the limited exciton diffusion length (ca. 10 ~20 nm) in organic materials and the large domain size of DOC8-IC, the excitons generated in the acceptor domain cannot be efficiently diffused to the donor-acceptor interface. This is why DOC8-IC shows the lowest PCE among these three NC-FREAs.”

Domain purity calculation needs to normalize with materials optical contrast between different donor and acceptor, which needs the synchrotron near-edge X-ray absorption fine structure (NEXAFS) spectra that are difficult to obtain in the short revision time. Thus, we tried our best to use miscibility to clarify the structure-performance relationship.

3. The miscibility was calculated by surface energy (see SI in Page S4). All information are provided in the revised manuscript. (Page 6)

Details are as follows:

“The surface energy γ values could be calculated according to the Wu model on the neat films by the Equation:[S4]

$$\gamma_{LV}(1 + \cos \theta) = \frac{4\gamma_S^d \gamma_L^d}{\gamma_S^d + \gamma_L^d} + \frac{4\gamma_S^p \gamma_L^p}{\gamma_S^p + \gamma_L^p}$$

$$\gamma = \gamma^d + \gamma^p$$

And the two different contact angles of water and glycerol were measured to achieve the γ of acceptor and polymer donor, as shown in **Table S5**. And the γ is the sum of dispersion (d) and polar (p) components.

Table S5. Contact angle of water and glycerol and surface tension of **DOC6-IC**, **DOC8-IC**, **DOC2C6-IC**, **DC6-IC**, **DOC2C6-2F**, and **PBDB-T**.

	θ_{water} [deg]	θ_{GL} [deg]	γ [mJ·m ⁻¹]
DOC6-IC	99.3	85.4	23.49
DOC8-IC	101.4	85.5	23.81
DOC2C6-IC	99.5	86	23.15
DC6-IC	98.2	91.1	21.77
DOC2C6-2F	99.3	86.6	22.85
PBDB-T	104.1	92	20.17

As $\delta \propto \sqrt{\gamma}$, we could calculate all the solubility parameter (δ) of acceptor and polymer.^[S5]

Further, the Flory-Huggins interaction parameter χ could be calculated according to the Equation:^[S5]

$$\chi_{11} = \frac{V_1}{RT} (\delta_i - \delta_j)^2 + 0.34$$

Since we adopted the *o*-DCB as the solvent, the V_1 is 113.3 cm³ mol⁻¹. We calculated the χ of the blend of small molecular acceptors and PBDB-T, and the results are shown in **Table S6**.

Table S6. Flory-Huggins Interaction Parameters for acceptors and donor.

Active layers	χ
DOC6-IC/PBDB-T	0.42
DOC8-IC/PBDB-T	0.44
DOC2C6-IC/PBDB-T	0.41
DC6-IC/PBDB-T	0.36
DOC2C6-2F/PBDB-T	0.39

“It has been pointed out that a low χ value denotes a high miscibility of blend films with a more homogeneous morphology,³⁵ which would benefit the charge transport and exciton dissociation in the active layer. **DOC8-IC** possesses the highest value of χ among these small molecular acceptors, namely it has the lowest miscibility with PBDB-T and the largest domain size, in accordance with the R-SoXS and fluorescence quenching measurements.”

- As the molecular weight of PBDB-T type conjugated polymers likely affects the device reproducibility and many characterizations (e.g., DOI: 10.1016/j.joule.2018.11.006; DOI: 10.1002/solr.201800129). The authors should add and discuss such information.

Answer:

We have provided the molecular weight of PBDB-T ($M_n=115,000$, PDI = 1.63) and added some discussions (Page 4). We cited the two references recommended by the reviewer in

the revised manuscript (Ref. 24, 25). Details are as follows:

“...the molecular weight of PBDB-T has a significant influence on the morphology of active layer,^{24,25} the one we used here is from the same batch, which has a M_n of 115,000 and a PDI of 1.63.”

Reviewer #3 (Remarks to the Author):

Most non-fullerene acceptors developed to date for high-performance organic solar cells feature a fused ladder-type electron-rich core. In this work, Bo and co-workers developed a series of non-fullerene acceptor without such fused core. The incorporation of alkoxy chain triggers intramolecular non-covalent S-O interaction, locking the acceptor materials backbone to achieve planar conformation. In addition, this dynamic non-covalent interaction can greatly reduce the synthetic complexity and hence the materials can be accessed at low-cost. The strong electron-donating ability of alkoxy chain markedly broadens the absorption to the near-infrared region and the planer backbone facilitates charge transport. When incorporated into photovoltaic devices, the optimized solar cells with PBDB-T:DOC2C6-2F active layer attained a remarkable power conversion efficiency (PCE) of >13%. Although a PCE of >15% has been achieved in non-fullerene solar cells very recently, this device performance is still highly promising in terms of the chemical structure novelty and materials design approach. Moreover, various materials and device characterization techniques were well performed, which help establish the structure-property correlations and offer insights into future materials design. In principle,

the manuscript is well-written and the data are supportive to the conclusion. Hence, publication of this manuscript is recommended in this journal after the following issues are addressed.

1) I am not sure if this series of materials can be called supramolecular electron acceptors since the non-covalent S-O interaction is intramolecular rather than intermolecular (or supramolecular).

Answer: Thanks for the reviewer's valuable comment. We have changed the description "supramolecular electron acceptors" to "noncovalently fused-ring electron acceptors" in the revised manuscript.

2) The study showed a low non-radiative recombination of energy loss in the range of 0.20 to 0.27 eV, the overall energy loss should be calculated and included.

Answer:

Thanks for the reviewer's nice suggestion. The overall energy loss has added in Table S4 of the supporting information (Page S3).

3) Please provide the more detailed information for intermediates, such as ¹³C NMR, element analyses and HRMS. Scheme 1. in vi) a) chloroform, reflux; b) chloroform, r.t., this is confusing. The structure of polymer donor should be included somewhere in the main text.

Answer:

Thanks for the nice advice. We have provided more characterizations for the compounds in the section “1.7 Synthesis” of supporting information, such as the carbon NMR spectra and MALDI-TOF data. And we replaced the caption of Scheme 1 in the revised manuscript (Page 2).

Details: “vi) for **6a**, chloroform, reflux; for **6b**, chloroform, r.t.”

And the structure of polymer donor PBDB-T was added in **Scheme 1b** (Page 2).

4) Figure 1b shows the absorbance coefficient of solution, while the absorption of film is normalized. The film absorption should also be based on absorbance coefficient since solar cell is film-based device. In addition, I would like to see UV-Vis of the blends-how does the absorption strength of these “supramolecular electron acceptors” compared to that of the PBDB-T?

Answer:

Thanks for the nice advice. We have provided the film absorption spectra with absorbance coefficient of both small molecules and PBDB-T in **Figure 1** of the revised manuscript (Page 3).

5) The non-covalent S-O interaction (or conformation lock) has been known for a long time, the first usage of this approach for developing high-performance organic semiconductors was pioneered by Prof. Mark D. Watson, which should be mentioned in the manuscript. In addition, two highly relevant works, J. Am. Chem. Soc. 2013, 135, 1986. and Adv. Mater. 2016, 28, 9969, which should be cited in appropriate places.

Answer:

Thanks for the nice advice. We have cited these references in the revised manuscript (Page 2 and Ref. 20, 21).

Details:

“The S...O interactions, which have been used in developing high-performance organic semiconductors pioneered by Watson et al., could endow the central donor core with a planar ladder-like structure.¹⁹⁻²¹”

6) On Page #1 “....., we for the first time developed a kind of supramolecular, the intramolecular noncovalent interactions in which locked the aromatic units in the molecular backbone to form supramolecular fused-ring structures.” Please re-phrase this sentence. On same page, “...two branched octyl side chains...” I don't think that you can call 2-ethylhexyl as a branched octyl chain.

Answer:

We have rephrased these sentences in the revised manuscript (Page 2).

Details:

“In our previous work,¹⁷ we for the first time developed a kind of noncovalently fused-ring electron acceptors (NC FREAs), in which the intramolecular noncovalent interactions can lock the aromatic units to form a ladder-like structure.”

“An as-cast solar cell with **DOC2C6-IC**, which bears two 2-ethylhexyl chains on the central phenyl unit, achieves a high PCE of 11.10%.”

7) *".....display broad structured absorption ranging from 500 to 900 nm with two peaks....". One should be absorption shoulder.*

Answer:

This sentence has been changed as follows:

"In films, the four NC FREAs with a central 2,5-bis(alkyloxy)phenylene unit display broad structured absorption ranging from 500 to 900 nm with a peak located at approximately 780 nm and a shoulder at approximately 710 nm." (Page 3).

8) *The J_{sc} values from $J-V$ curves should be included in Table 1.*

Answer:

The J_{sc} values obtained both from $J-V$ curves and the integration of EQE curves have been included in Table 1.

9) *From DOC6-IC to DOC8-IC and to DOC2C6-IC, the chain becomes more bulkier, however the device performance didn't show a monotonic increasing trend, please comment on this. Particularly, the image between film morphology and device performance is still not very clear. The RoSX can provide information on domain size and purity, can author include these data?*

Answer:

Thanks for the reviewer's suggestion. We have noticed the unusual trend of the PCE among DOC6-IC, DOC8-IC, and DOC2C6-IC, so we carried out the fluorescence quenching measurement. The DOC8-IC showed the worst quenching with PBDB-T

among these acceptors. (Page 4) In addition, to better understand the relationship between morphology and PCE, we further carried out R-SoXS experiments to analyze the morphology of active layers. And the information is added in Page 4 and Figure 3d.

Details:

“Fluorescence quenching experiments were also employed to evaluate the domain size of each component in the blend films.²² Fluorescence spectra of neat small-molecule acceptor films, neat PBDB-T film, and blend films are shown in Figure S6.”

The quenching of fluorescence in the DOC8-IC:PBDB-T blend film is much poorer than that in the other two blend films, indicating that the domain size of acceptor in the DOC8-IC:PBDB-T blend film is the biggest among these three blend films.

The domain size was investigated by R-SoXS. Due to the difficulty on obtaining synchrotron NEXAFS as mentioned above, the domain purity is not included in the current version.

Details are as follows:” We find that the active layer of DOC8-IC shows larger domain size of 25.7 nm, while those of DOC6-IC and DOC2C6-IC exhibit smaller domain size of 22.7 nm and 15.7 nm, respectively. Due to the limited exciton diffusion length (ca. 10 ~20 nm) in organic materials and the large domain size of DOC8-IC, the excitons generated in the acceptor domain cannot be efficiently diffused to the donor-acceptor interface. This is why DOC8-IC shows the lowest PCE among these three NC FREAs.” Regarding to these three acceptors, the domain size is closely related to the device performance.

10) *The authors claimed that the all blend films are highly smooth, the specific data should be included.*

Answer:

We have provided the RMS roughness data in the revised manuscript (Page 4).

Details:

“Atomic force microscope (AFM) images reveal that the surfaces of all blend films are highly smooth (Figure S7). The root-mean-square (RMS) roughness values of **DOC6-IC**, **DOC8-IC**, **DOC2C6-IC**, **DC6-IC** and **DOC2C6-2F** based blend films are 2.45, 2.55, 2.70, 3.32, and 1.16 nm, respectively.”

11) *“...Meanwhile the outstanding (010) peak in the OOP direction shows an enhanced face-on orientation of DOC2C6-2F.” The driving force for such microstructural feature?*

Answer:

Thanks the reviewer’s nice suggestion. The possible driving force for such microstructural feature might be due to the introduction of the fluorine atoms to the terminal groups of DOC2C6-2F, which can enhance intermolecular D-A interaction (also called as electrostatic interaction) between neighboring acceptor molecules due to the strong electron withdrawing ability of fluorine atom (*Chem. Mater.* 2018, 30, 4307). This can improve the crystallization of the molecule which is in accordance with the result of DSC.

Some descriptions have been added in the revised manuscript (Page 6).

Details:

“Meanwhile the outstanding (010) peak in the OOP direction shows an enhanced face-on orientation of **DOC2C6-2F**. The possible driving force for such microstructural feature might be due to the introduction of fluorine atoms at the terminal groups of **DOC2C6-2F**, which can enhance intermolecular D-A interaction (also called as electrostatic interaction) between neighboring acceptor molecules due to the strong electron withdrawing ability of fluorine atom.²⁹ This can improve the crystallization of molecules, in accordance with the result of DSC. ”

12) More background for transfer matrix simulation should be provided, particularly how the maximum photocurrent was derived.

Answer:

Thanks the reviewer’s nice suggestion. We have added more discussions on how transfer matrix simulation was done, and how the maximum photocurrent was derived (Page 6).

The transfer matrix simulation is a standard optical simulation tool for thin film optics, and it was firstly used for organic solar cells in 1999 (Ref. 36).

“Using TM simulated active layer absorption, the maximum photocurrent extraction of the solar cells based on NC FREAs and DC6-IC is calculated for different active layer thicknesses, by assuming 100% device internal quantum efficiency (IQE).”

13) There are a new series of electron acceptors, the mobilities in neat films should be also measured by the SCLC method.

Answer:

The mobilities in neat films have been provided in the supporting information (Table S10 and Figure S11 in Page S22). And the trend of neat small molecular acceptor films is consistent with that of blend films.

Reviewer #4 (Remarks to the Author):

This manuscript reports a series of “supramolecular electron acceptors” (SMEAs) containing alkoxy side chains and dicyanoindanone end groups are synthesized, characterized, and used for the fabrication of solar cells. Compared to the more commonly employed fused-ring electron acceptors (FREAs), the authors claim that the SMEAs are easy and cheap to synthesize, by avoiding the need to build a system of fused rings. In addition, the introduction of the alkoxy side in the donor core is postulated to lead to an S-O conformation locking interaction, reinforcing a planar conformation that is crucial for π -electron delocalization. Here, a variety of techniques to study the neat material and blends with the polymer PBDB-T, including absorption spectroscopy, GIWAXS, AFM, TEM, photophysical measurements, and SCLC mobility to study their molecules and system. Through their characterization, the authors found that the SMEA DOC2C6-2F exhibited the highest PCE of the molecules under study, although the reason for this is not the focus of the manuscript. Instead, the authors choose to focus on how the SMEAs differ from a control “non-SMEA” (referred to as DC6-IC) that is nearly identical to the SMEAs, except that the core alkoxy side chains have been replaced with a hexyl side chain. The authors find that compared to the SMEAs, DC6-IC has a blue shifted, more narrow

absorption profile, greater non-radiative recombination losses, and lower FF and Jsc.

This work could be suitable for publication after addressing the points below.

Specific Comments

1) The authors make the claim several times throughout the manuscript that synthesis of SMEAs is simple and cheap compared to that of FREAs. However, no evidence or cost analysis of this is provided. In fact, compared to ITIC-4F, a well-known acceptor that performs comparably to the best molecule presented here, the synthetic sequence and cost appear similar at best. To make the core of ITIC-4F, thienothiophene is coupled to 2,5-dibromoterephthalic acid diethylester (which is commercially available and can be derived from the corresponding acid, which is available for <\$30/10 g) by Negishi coupling. The alkyl chains are then added in, and the compound is cyclized with acid in 1 pot. The remainder of the syntheses are identical: formylation followed by Knoevenagel condensation of the same difluorinated dicyanoindanone. This is a total of 4 steps, or 5 if the addition and cyclization are performed separately. In the synthesis outlined for the SMEAs, 4 steps are required. In addition, the cyclopentadithiophene compound used for the Stille coupling (which itself is extremely costly due to the use of toxic tin compounds and should be replaced with a Negishi reaction) is ~4x as expensive as thienothiophene (\$400/g vs \$103/g from sigma aldrich) which is used for the synthesis of most FREAs. This difference in cost reflects the long sequence required for the synthesis of cyclopentadithiophene compounds relative to thienothiophene. If the authors wish to keep

the claim that SMEAs are cheap and easy to make, they need to consider not only the step count from commercially available components, but the cost and synthetic challenges associated with the components they are using.

Answer:

Thanks for the nice comment. The synthesis of such NC FREAs may be not low-cost in comparison with ITIC-4F, since ITIC-4F contains 7-fused rings and these NC FREAs comprise 9-quasi fused rings. In addition, DOC2C6-2F can afford a high efficiency of 13.24% when blended with PBDB-T; whereas ITIC-4F can give PCEs of 13.1% and 9.34% when blended with PBDB-T-SF (DOI: 10.1021/jacs.7b02677) and PBDB-T (DOI: 10.1002/ange.201807865), respectively. We should note that the synthesis of PBDB-T-SF is more complicated than PBDB-T.

To illustrate that the synthesis of noncovalently fused-ring acceptor (A1) is cheaper than that of the corresponding fused-ring acceptor (A2), we outlined their synthetic routes in Scheme A. Starting from compound a, it requires three steps to achieve the aimed acceptor A1 as shown in route a. Each step can be accomplished in relatively high yield. As for A2, two extra steps are required to obtain covalently fused structure as shown in route b. According to the literature, the total yield of these two extra steps is usually low (ca. 30% to 50%).

Scheme A

2) Although not clearly stated, the authors make a distinction between DC6-IC (non-SMEA) and the all the other molecules (SMEAs) in the manuscript on the basis that DC6-IC does not contain S-O interactions that lock the planarity of the donor core into place. This reviewer takes issue with this distinction. All molecule reported in this manuscript almost certainly contain S-O conformation locking interactions between the dicyanoindanone end group and the cyclopentadithiophene core group. The chemical structures in scheme 1 should be redrawn to reflect this.

Answer:

Thanks for the nice advice. We agree with the reviewer that the S-O conformation locking interactions between the dicyanoindanone end group and the cyclopentadithiophene core group actually exists, but in our work we emphasized the S-O interaction in the core unit:

“the ladder-like core is composed of a central 2,5-bis(alkyloxy)phenylene unit flanked by two 4,4-bis(2-ethylhexyl)-4H-cyclopenta[2,1-b:3,4-b']dithiophene units. The S-O interactions, which have been used in developing high-performance organic semiconductors pioneered by Watson et al., could endow the central donor unit with a planar ladder-like structure.” And we have provided the structures in Scheme 1 in the revised manuscript to make clear the S-O interactions in the core unit to avoid misunderstanding.

Furthermore, conformation locking interaction have been reported in numerous polymers and small molecule semiconductors (review: DOI: 10.1021/acs.chemrev.7b00084). More to the point, S-O conformation interactions have been directly observed in one subgroup of FREA crystal structures (DOI: <https://doi.org/10.1073/pnas.1807535115>), and is undoubtedly present in many others. Should these molecules also be considered SMEAs because of the conformation locking non-covalent interactions?

Answer:

Thanks for the reviewer’s nice comment. It is true that conformation locking interaction has been well reported in many polymers and small molecular semiconductors. To the best of our knowledge, the S-O interaction was directly observed in the single crystal structures by Reynolds et al. (*J. Polym. Sci., Part A: Polym. Chem.* 2001, 39, 2164). In our early paper, we also used S-O intramolecular interaction in the synthesis of polymer donors (*Macromolecules* 2012, 45, 7843-7854). Although S-O interaction existed in many high efficiency FREAs, we first pointed out that intramolecular noncovalent (secondary)

interactions play a pivotal role to the photovoltaic performance (*J. Am. Chem. Soc.* 2017, 139, 3356-3359). We think the S-O interaction can lock the end group and the central fused-ring core to make the FREA molecule take on a planar conformation, which endows the molecule with better conjugation. Here we only want to emphasize that besides the S-O interaction in the terminal group, the use of intramolecular secondary interactions (including S-O interaction) in the donor core of A-D-A type acceptors not only can give good photovoltaic performance, but also can lower the synthetic cost. We want to emphasize the importance of using noncovalent interaction in the designing of high efficiency electron acceptors.

3) *In the fluorescence quenching experiments, the exact percentage of fluorescence quenching should be determined.*

Answer:

We have provided such information in the supporting information Figure S6.

4) *Based on the fluorescence quenching experiments, the authors propose that DOC8-IC forms much larger domains than the others, as the quenching of the DOC8-IC by PBDB-T is much less than the others. This is plausible, but the TEM images seem to show no difference and the CLs measured by GIWAXS are fairly similar between DOC8-IC and the others. RSoXS or another technique should be performed to clearly measure the domain size distribution in the blends.*

Answer:

Thanks for the reviewer's suggestion. To better understand the relationship between morphology and PCE, we further carried out R-SoXS experiments to analyze the morphology of active layers. And the information is added in Page 4 and Figure 3d.

Details:

"We find that the active layer of DOC8-IC shows larger domain size of 25.7 nm, while those of DOC6-IC and DOC2C6-IC exhibit smaller domain size of 22.7 nm and 15.7 nm, respectively. Due to the limited exciton diffusion length (ca. 10 ~20 nm) in organic materials and the large domain size of DOC8-IC, the excitons generated in the acceptor domain cannot be efficiently diffused to the donor-acceptor interface. This is why DOC8-IC shows the lowest PCE among these three NC FREAs."

5) The GIWAXS images are small and of low resolution. They are hard to read, and when zooming in the images become blurry.

Answer:

Thanks for the advice. We have replaced these images in the revised manuscript. (Figure 3, Page 5)

6) The authors state that the molecules DOC8-IC and DC6-IC exhibit no preferred orientation on the basis of the GIWAXS data. Does this mean isotropic distribution?

Answer:

The GIWAXS data show that no preferred orientation is observed for DOC8-IC and DC6-IC in the neat film, but they tend to adopt a face-on orientation in the blend films with

PBDB-T. We can say the molecules DOC8-IC and DC6-IC themselves prefer to be isotropic distribution. Fortunately, in the blend films DOC8-IC and DC6-IC show some preferred face-on orientation (Page 6).

7) *FTPS-EQE should be defined when first used.*

Answer:

Thanks for the reviewer's suggestion. We have defined FTPS-EQE in the revised manuscript (Page 7).

Details: "Such a low energy loss is critically important for achieving high V_{oc} , and it is also confirmed by a sensitive Fourier Transform Photocurrent Spectroscopy-EQE (FTPS-EQE) measurement.³⁸"

REVIEWERS' COMMENTS:

Reviewer #2 (Remarks to the Author):

The manuscript "Noncovalently Fused-Ring Electron Acceptors with Near-Infrared Absorption for High-Performance Organic Solar Cells with Low Voltage Losses" by Bo and coworkers has undergone substantial revisions, which I believe have clarified most of the questions raised in the course of its review.

To address the major concerns of four reviewers, they made the following the key improvements:

- 1) Improved the terminology used in the paper: "supramolecular electron acceptors" was replaced with "noncovalently fused-ring electron acceptors"
- 2) Added descriptions about the cost of new acceptors by outlining their synthetic routes in Scheme A.
- 3) Included new supportive data, covering a multitude of in-depth analysis: performed TOF-SIMS experiments to study the vertical composition profiles (Figure S9) and RSoXS experiments to analyze the domain size distributions (Figure 3d). Surface energy measurements are used to estimate the miscibility between donors and acceptors.
- 4) Reported SCLC mobility for new acceptors in supporting information Table S10 and Figure S11 and RMS roughness data in Page 4 and Urbach energy in Supplementary Information Table S4
- 5) Provided more detailed information for the donor polymer and intermediates, such as ¹³C NMR, TOF, GPC data.

Overall, these revisions provided a clear mechanistic explanation of the difference between the acceptors. I am convinced that the authors have made substantial changes to address the concerns of all reviewers and the data are now supportive of the conclusion. I am thus happy to recommend it for publication in Nature Communications.

A minor point:

In response to my prior comment "The rules by which bandgap has been evaluated needs to be articulated and put in the context of the literature", the authors stated that band gaps were not needed for analyzing voltage losses. But why?

Reviewer #3 (Remarks to the Author):

The authors gave the credits to Mark D. Watson for his pioneering work on S-O non-covalent interaction, however, none of references 19-21 is from Watson. The reference 19 should be replaced by J. Am. Chem. Soc., 2009, 131 (21), 7206–7207. Except for this, the authors have well addressed my other concerns.

After this, the manuscript should be published in Nature Communications.

Reviewer #4 (Remarks to the Author):

The authors took very seriously all of the suggestions by the reviewers and have address most of the comments from them. I believe the quality of the work has greatly improved and feel that this paper can be published now.

REVIEWERS' COMMENTS:

Reviewer #2 (Remarks to the Author):

The manuscript “Noncovalently Fused-Ring Electron Acceptors with Near-Infrared Absorption for High-Performance Organic Solar Cells with Low Voltage Losses” by Bo and coworkers has undergone substantial revisions, which I believe have clarified most of the questions raised in the course of its review.

To address the major concerns of four reviewers, they made the following the key improvements:

- 1) Improved the terminology used in the paper: “supramolecular electron acceptors” was replaced with “noncovalently fused-ring electron acceptors”*
- 2) Added descriptions about the cost of new acceptors by outlining their synthetic routes in Scheme A.*
- 3) Included new supportive data, covering a multitude of in-depth analysis: performed TOF-SIMS experiments to study the vertical composition profiles (Figure S9) and RSoXS experiments to analyze the domain size distributions (Figure 3d). Surface energy measurements are used to estimate the miscibility between donors and acceptors.*
- 4) Reported SCLC mobility for new acceptors in supporting information Table S10 and Figure S11 and RMS roughness data in Page 4 and Urbach energy in Supplementary Information Table S4*
- 5) Provided more detailed information for the donor polymer and intermediates, such as ¹³C NMR, TOF, GPC data.*

Overall, these revisions provided a clear mechanistic explanation of the difference between the acceptors. I am convinced that the authors have made substantial changes to address the concerns of all reviewers and the data are now supportive of the conclusion. I am thus happy to recommend it for publication in Nature Communications.

A minor point:

In response to my prior comment “The rules by which bandgap has been evaluated needs to be articulated and put in the context of the literature”, the authors stated that band gaps were not needed for analyzing voltage losses. But why?

Answer:

In this manuscript, we mainly discussed voltage losses due to non-radiative recombination of charge carriers. Non-radiative recombination loss is directly related to the experimentally determined electroluminescence external quantum efficiency of the solar cell devices. This is why we stated that we did not need the numerical value of the bandgap to analyze voltage losses in this manuscript. As described in the section ‘**Voltage losses in the solar cells based on NC-FREAs**’ in the manuscript, ‘The second term does not depend on E_g , but on the external quantum efficiency of electroluminescence (EQE_{EL}) of the solar cell and it denotes the V_{oc} loss is induced by non-radiative recombination of charge carriers. It should be noted that EQE_{EL} can be experimentally determined.’

Reviewer #3 (Remarks to the Author):

The authors gave the credits to Mark D. Watson for his pioneering work on S-O non-covalent interaction, however, none of references 19-21 is from Watson. The reference 19 should be replaced by J. Am. Chem. Soc., 2009, 131 (21), 7206–7207. Except for this, the authors have well addressed my other concerns.

Answer:

Thanks for the reviewer's advice. We have replaced the reference 19 to the work of Mark D. Watson.

After this, the manuscript should be published in Nature Communications.

Reviewer #4 (Remarks to the Author):

The authors took very seriously all of the suggestions by the reviewers and have address most of the comments from them. I believe the quality of the work has greatly improved and feel that this paper can be published now.